# Highly host-linked viromes in the built environment possess habitat-dependent diversity and functions for potential virus-host coevolution

Shicong Du [1], Xinzhao Tong[1,2], Alvin C. K. Lai[1], Chak K. Chan[1], Christopher E. Mason [3,4,5,6] & Patrick K. H. Lee [1,7] ✉

Viruses in built environments (BEs) raise public health concerns, yet they are generally less studied than bacteria. To better understand viral dynamics in BEs, this study assesses viromes from 11 habitats across four types of BEs with low to high occupancy. The diversity, composition, metabolic functions, and lifestyles of the viromes are found to be habitat dependent. Caudoviricetes species are ubiquitous on surface habitats in the BEs, and some of them are distinct from those present in other environments. Antimicrobial resistance genes are identified in viruses inhabiting surfaces frequently touched by occupants and in viruses inhabiting occupants' skin. Diverse CRISPR/Cas immunity systems and anti-CRISPR proteins are found in bacterial hosts and viruses, respectively, consistent with the strongly coupled virus–host links. Evidence of viruses potentially aiding host adaptation in a specific-habitat manner is identified through a unique gene insertion. This work illustrates that virus–host interactions occur frequently in BEs and that viruses are integral members of BE microbiomes.

Viruses warrant our attention because they have potentially detrimental impacts on human health[1] but also play crucial roles in many ecosystems[2–4]. Built environments (BEs), where people typically spend most of their lives, harbor a rich diversity of microorganisms[5], but most studies of BEs have largely focused on bacteria and fungi while overlooking viruses[6,7]. The total concentration of the viruses in BEs is estimated to be ~$10^5$ particles/cubic meter[8]. Although the environmental conditions of most BEs are oligotrophic and considered poorly suited for microbial life[9], a conspicuous diversity of viruses, including epidemic-associated viruses (e.g., SARS-CoV-2[10] and yellow fever virus[11]), have been found in microbial communities in air and on surfaces in BEs. A few studies on viromes in public buildings (e.g., daycare centers and restrooms) have mainly focused on a small spatial scale and limited sample types and have not investigated the bacterial hosts of the viruses[12–14]. A recent global-scale study that applied bulk metagenomic sequencing without virus enrichment provided evidence that viruses are ubiquitous on public surfaces in BEs[15].

Virus–host interactions are central to the ecology and evolution of microbiomes in diverse ecosystems[4,16,17]. Recent advances in bioinformatic tools have enabled accurate prediction of the association between metagenome-derived viruses and their potential bacterial hosts, including exact matches of molecular signals (namely clustered

[1]School of Energy and Environment, City University of Hong Kong, Hong Kong SAR, China. [2]Department of Biological Sciences, School of Science, Xi'an Jiaotong-Liverpool University, Suzhou, P. R. China. [3]Department of Physiology and Biophysics, Weill Cornell Medicine, New York, NY, USA. [4]The HRH Prince Alwaleed Bin Talal Bin Abdulaziz Alsaud Institute for Computational Biomedicine, Weill Cornell Medicine, New York, NY, USA. [5]The WorldQuant Initiative for Quantitative Prediction, Weill Cornell Medicine, New York, NY, USA. [6]The Feil Family Brain and Mind Research Institute, Weill Cornell Medicine, New York, NY, USA. [7]State Key Laboratory of Marine Pollution, City University of Hong Kong, Hong Kong SAR, China. ✉e-mail: patrick.kh.lee@cityu.edu.hk

regularly interspaced short palindromic repeat [CRISPR] spacer, integrated genome, and tRNA) and consistent k-mer frequency[18]. Phages have evolved diverse lifestyle and transmission strategies, such as temperate–lytic life cycle switching, transduction, and host gene disruption, to exploit the hosts' cellular machinery for reproduction[19]. In most marine and soil environments, phages are often highly diverse and abundant, thereby routinely infecting a significant fraction of their microbial hosts, which, together with the expression of virus-encoded auxiliary metabolic genes (AMGs) in host genomes, plays a key role in global nutrient cycling[4,20,21]. From an ecological perspective, phages in a microbial community can mediate the competition among bacterial species by establishing lytic infections through several well-established ecological models, including the "kill-the-winner" and "piggyback-the-winner" models[22].

While phages can drive rapid genetic and phenotypic changes in bacteria, bacterial hosts can also readily evolve defense mechanisms to counter phage attacks through de novo mutation and other molecular mechanisms[23]. Recently, various functional CRISPR/CRISPR-associated (Cas) systems in bacteria have been identified in a body-wide human metagenomic study[24]. However, to antagonize the host immune system, phages have evolved anti-CRISPR (Acr) proteins to inactivate bacterial Cas nucleases during infection[25]. Long-term inactivation of CRISPR/Cas by inhibitor phages can lead to the loss and even absence of CRISPR/Cas in some bacterial lineages[26].

CRISPR/Cas systems have been reported in surface microbiomes across urban environments worldwide[15]; however, the immune mechanisms of infection and the virus–host interactions (e.g., the extent of virus–host links, the prevalent viral life cycle, and the novelty of Acr proteins) that occur in BEs are poorly understood. To fill this knowledge gap and explore the diversity and ecosystem functions of viruses in BEs, 738 bulk metagenomes from diverse habitats across different BEs in Hong Kong (HK) were investigated in this study. The highly coupled virus–host interactions identified in this study support the notion that viruses aid the adaptation of bacterial hosts to the specific environmental conditions of BEs and that the abundance of most bacterial populations in BEs is strongly correlated with their resident viruses. This study provides evidence that viruses are integral members of BE microbiomes.

## Results

### Habitat-dependent diversity and distribution of the BE viromes

From the 738 bulk metagenomes collected from rural and urban BEs in HK, including piers, public facilities, residences, and subways (Fig. S1a, Supplementary Data 1), ~4.5 million assembled contigs were generated with MetaWRAP (see "Methods"), with lengths mostly between 1 and 3 kb were obtained (Fig. S1b). Viral contigs were identified using Visorter2[27] and DeepVirFinder[28]; the latter showed a better performance for shorter contigs (1–3 kb; Fig. S1c). In total, 594,851 unique viral contigs with lengths ≥1 kb were recovered from all samples (Fig. 1a). After quality filtering, 1174 viral genomes with completeness ≥50% (98 complete, 346 high-quality, and 730 medium-quality genomes) were identified (Fig. S1d, Supplementary Data 2). These genomes were well represented across the four types of BEs (Fig. 1b), with 66% of them detected on surfaces in residences (Fig. 1c). Despite analyzing the bulk metagenomes, only 28% of the viral genomes showed evidence of host integration based on an assessment of the provirus integration sites (i.e., the host region was predicted on both ends of viral genomes) (Fig. 1c). Of the 471 viral operational taxonomic units (vOTUs) identified, 355 were found in at least two samples (Fig. S2a); among the types of BE, the largest number of vOTUs were found in residences. At a higher taxonomic ranking, the viral genomes were clustered into 332 and 220 genus- and family-level vOTUs, respectively. The rarefaction curves of the vOTUs did not reach a plateau, suggesting that additional samples are required to capture the virome diversity in BEs within a city (Fig. S2a).

Given that the samples were collected from different habitats in terms of sources (i.e., air and surface) and materials (i.e., concrete, metal, and wood), the habitat-dependent features of the viromes were further investigated. Viral genomes were recovered most frequently from occupants' skin (21% of all samples) and doorknobs (15%) and least frequently from air (2%) (Fig. S2b). The virome composition mostly differed between habitats (analysis of similarity R = 0.355, $p < 0.001$), and permutational multivariate analysis of variance confirmed that habitat was the main driver of variation ($R^2 = 0.148$, $p < 0.001$) (Fig. 1d). The airborne virome was distinct from the surface-borne viromes, with a low within-habitat variance according to the Bray–Curtis dissimilarity distance in the principal coordinate analysis

**Fig. 1 | Annotation of the high-quality viral genomes recovered from metagenomes collected from built environments. a** Boxplots of the contig lengths of the predicted viral and non-viral contigs >1 kb, as determined by Virsorter2 (Vs2) and DeepVirFinder (DVF) and assessed by CheckV. The number of contigs (*n*) is indicated. Boxplots represent the median, the first quartiles and third quartiles with whiskers drawn within the 1.5 interquartile range value. Points outside the whiskers are outliers. **b** Accumulation curves of the viral genomes in the combined, pier, public facility, residence, and subway datasets. **c** Metadata and taxonomy of 1174 viral genomes with >50% completeness. **d** Principal coordinate analysis of the Bray–Curtis dissimilarity matrix for all of the samples. The color and shape of the symbols indicate the built environments and surface materials, respectively.

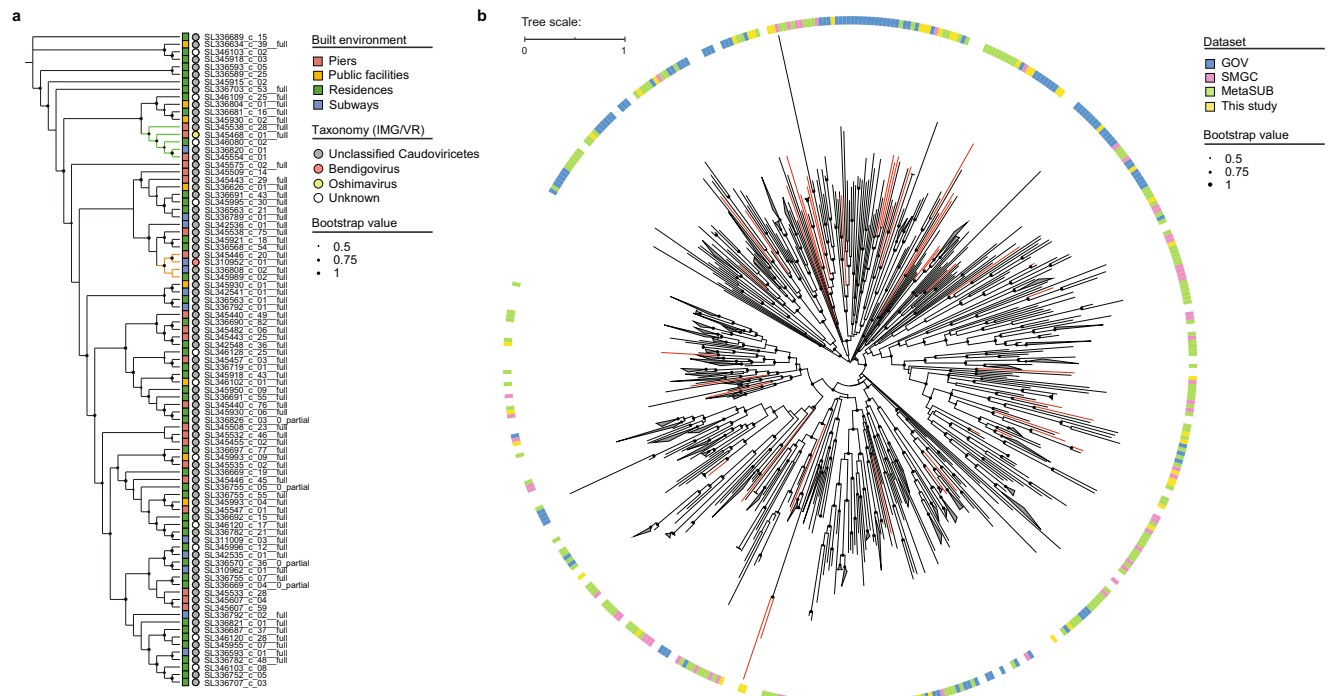

**Fig. 2 | Phylogenomics of the viral operational taxonomic units (vOTUs) from the class Caudoviricetes in the built environments (BEs). a** A maximum-likelihood phylogenetic tree of 87 species-level vOTUs derived from BEs. The branches of the two clusters of *Oshimavirus* and *Bendigovirus* are shown in light green and orange, respectively. The gray and white circles denote the novel Caudoviricetes viruses. **b** A maximum-likelihood phylogenetic tree of 599 species-level vOTUs derived from BEs and other datasets (SMGC, GOV, and MetaSUB). Branches of the novel Caudoviricetes vOTUs are shown in red. Lineages with branch lengths <0.5 kb were collapsed into a clade and are shown in white on the outer ring.

and the homogeneity of multivariate dispersions (permutational analysis of multivariate dispersions F = 53.29, *p* < 0.001) (Fig. 1d, S2c). Additionally, the vOTUs on handrails, poles, and ticket kiosks exhibited a significantly lower richness and Shannon diversity index than the vOTUs on occupants' skin and frequently touched indoor surfaces (e.g., doorknobs) (analysis of variance [ANOVA], *p* < 0.05; Fig. S2d, Supplementary Data 3). The species evenness varied significantly between habitats (ANOVA, *p* < 0.05), but the average evenness values of all habitats were > 0.85 (Fig. S2d), suggesting that no habitat had dominant vOTUs.

Next, the vOTUs were assigned to taxonomic ranks based on comparison with known viral sequences from the Integrated Microbial Genome and Viral (IMG/VR) database[29]. Most of the vOTUs (92.4%) could not be taxonomically classified into a known viral genus or family, similar to the reported rate of novelty for the viromes collected from other ecosystems[2,30], and could only be resolved as unclassified members of the class Caudoviricetes (Fig. 1c). Among the annotated vOTUs, 1.7%, 1.7%, 1.3%, and 0.6% belonged to the dsDNA viral genus *Pahexavirus*, ssRNA-RT viral family Metaviridae, ssDNA viral family Genomoviridae, and ssDNA viral family Inoviridae, respectively (Fig. S3). Specifically, the members of the family Autographiviridae, an extensively studied family of virulent phage[31], were enriched and dominant in subway air (Fig. S3, Supplementary Data 4); in contrast, the members of the genus *Pahexavirus* were abundant on doorknobs and skin surfaces (Fig. S3, Supplementary Data 4), which is not surprising because these have been shown to infect skin bacteria (e.g., Propionibacterium[32]). Furthermore, the members of the order Orthopolintovirales, an emerging group of viruses known as virophages[33], were also enriched on human skin-associated surfaces (Fig. S3). Notably, the human-associated papillomaviruses in the family Papillomaviridae, which can be transmitted directly or indirectly via skin contact[34], were mainly found in frequently touched habitats (i.e., doorknobs, ticket kiosks, and handrails in public facilities) (Fig. S3a).

## vOTUs of specific Caudoviricetes were selected from the BEs

Caudoviricetes, a class of viruses with a helical tail and icosahedral capsid (tailed bacteriophages), is prevalent in diverse ecosystems[2,30]. To investigate whether the members of Caudoviricetes that were present across the BE habitats possess a common evolutionary origin, a phylogenetic tree containing 87 species-level vOTUs was constructed using 77 reference protein-coding marker genes (Fig. 2a). The tree showed that the Caudoviricetes vOTUs that were widely distributed in different BE habitats but could not be classified through the IMG/VR database belonged to the genus *Bendigovirus*, whereas a distinct clade comprising one unknown vOTU belonged to the genus *Oshimavirus* (Fig. 2a).

The phylogeny of the Caudoviricetes members from the HK BEs was further analyzed against three other datasets of viromes, namely skin metagenomes primarily from subjects in North America (the Skin Microbial Genome Collection [SMGC] dataset)[35], the Global Ocean Viromes [GOV] dataset[36], and the Metagenomics and Metadesign of the Subways and Urban Biomes [MetaSUB] dataset[15]. As expected, the viral genomes derived from the HK BEs recruited 18.9% of the sequencing reads from this study, which was significantly higher than 0.3% and 0% recruited from the SMGC and GOV datasets, respectively (Fig. S4a). The GOV dataset result (0%) indicates that viruses in the marine environment are from different lineages than those in the BEs, which are under strong anthropogenic influences (Fig. S4a). A phylogenetic tree was further constructed with 599 species-level vOTUs from the four datasets; it showed that the vOTUs from the HK BEs were phylogenetically closer to those in the SMGC and MetaSUB datasets than to those in the GOV dataset (Fig. 2b). Based on the branch lengths, the vOTUs from the HK BEs and marine water had comparably high phylogenetic diversity (Fig. S4b). Nevertheless, some of the vOTUs from the HK BEs were clustered together (Fig. 2b), suggesting that the BE habitats were selective for certain vOTUs.

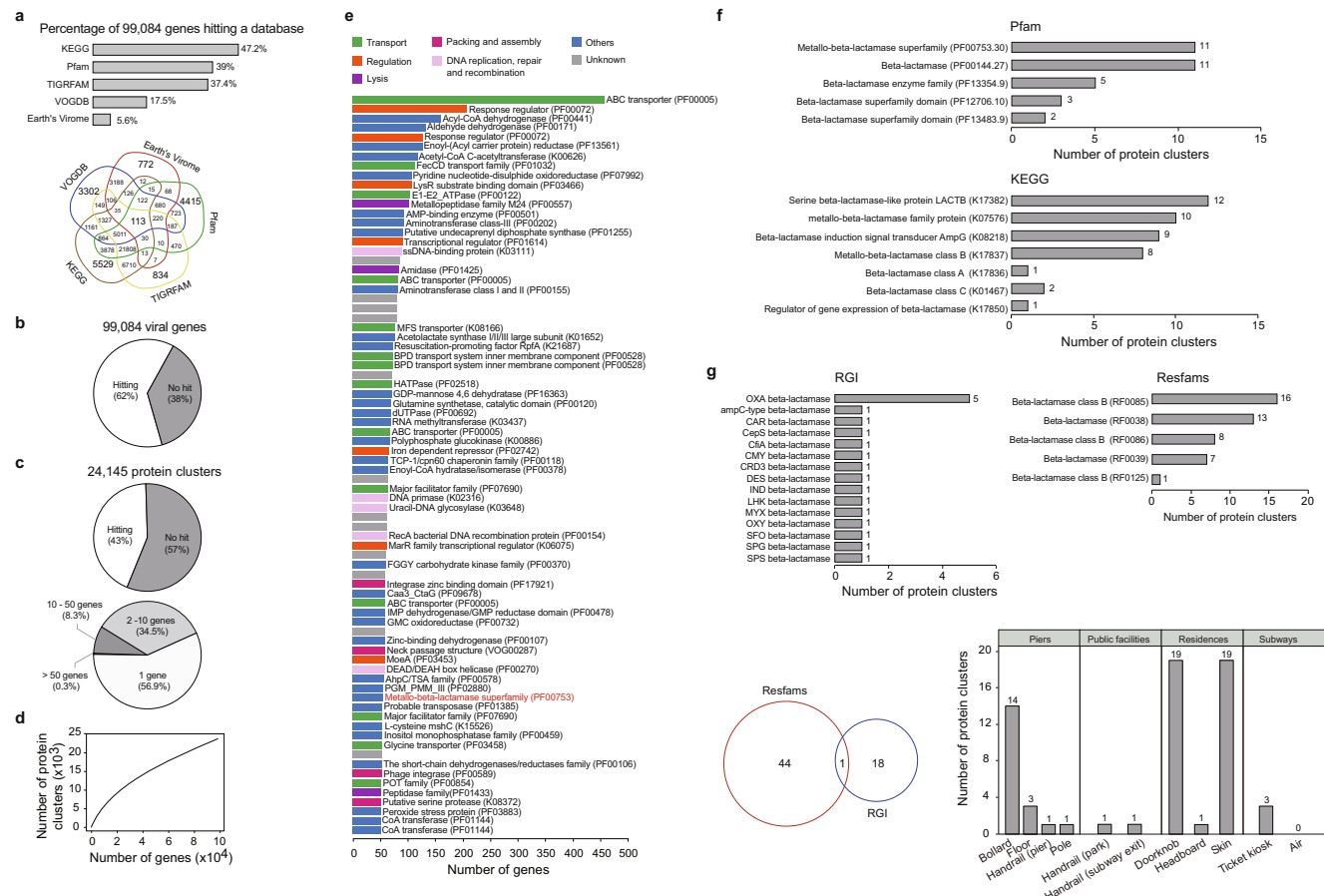

**Fig. 3 | Functional landscape of the built environment (BE) viromes. a** The percentage and number of protein-coding viral genes identified and shared among the five databases. **b** Percentage of genes that could or could not be matched to any database. **c** Percentage of protein-coding gene clusters that could or could not be matched to any database, and size distribution of protein-coding gene clusters based on the number of genes. **d** An accumulation curve of the protein-coding gene clusters. **e** Functional annotation of the protein-coding gene clusters with >50 genes. The protein-coding gene cluster that encodes beta-lactamase is highlighted in red. **f** The number of viral genes with putative beta-lactamase domains based on the Pfam and Kyoto Encyclopedia of Genes and Genomes (KEGG) databases. **g** The number of antimicrobial resistance genes (ARGs) encoding beta-lactamases based on the Resistance Gene Identifier (RGI) and Resfams databases. The Venn diagram summarizes the number of identified ARGs in the two databases, and the bar plot indicates the distribution of ARGs across BE habitats.

## Metabolic functions and antibiotic resistance genes (ARGs) encoded by the viromes

To explore the potential functional roles that viromes play in the BE microbiomes of HK, 99,084 protein-coding genes identified across the viromes were annotated using several databases. The results showed that 38% of the protein-coding genes had no significant database match, and ~2% were not assigned any biological functions (Fig. 3a, b), suggesting that little is known regarding the potential functions of viromes found in BEs. To further identify the shared functions among the viromes, all protein-coding genes were clustered into 24,145 de novo protein-coding gene clusters, and 43.1% of these clusters had at least two genes (Fig. 3c). The accumulation curve of protein-coding gene clusters was unsaturated, indicating that the collected samples exhibit a large diversity of functions (Fig. 3d). The largest protein-coding gene clusters with >50 genes mostly encoded proteins with functions in membrane transport (i.e., ABC transporter) and direct/indirect transcriptional regulation to control gene expression, genome replication, and transmission to other host cells (i.e., response regulator) (Fig. 3e). Other common viral functions, such as packaging, assembly, and lysis, were also found in the largest clusters (Fig. 3e).

While studies on gut viromes have shown that phages rarely encode ARGs[30, 37], it is unclear whether this characteristic is also common to viruses in BEs. Based on hidden Markov model searches against the Pfam[38] and Kyoto Encyclopedia of Genes and Genomes (KEGG)

databases[39], 53 unique protein-coding gene clusters that contained putative beta-lactamase-encoding genes were identified (Fig. 3f), and most of the ARGs were retrieved from viruses inhabiting human skin. The largest protein-coding gene cluster contained 55 genes encoding the metallo-beta-lactamases superfamily (PF00753) and genes encoding the LysR family of transcriptional regulators (K17850) (Fig. 3e). To further validate the antibiotic resistance capability of the BE viromes, a homology search against the curated ARG databases was performed using Resfams[40], NCBI AMRFinderPlus[41], and the Resistance Gene Identifier (RGI)[42]. Sixty-three beta-lactamase-coding gene clusters were identified, including 45 from Resfams and 19 from the RGI but none from AMRFinder, and these clusters were mainly distributed on bollard, doorknob, and skin surfaces (Fig. 3g). The search against Resfams also identified other potential ARGs, including *erm*AC, *van*ABCDHRXZ, and *tet*ADEHY (Fig. S5), which confer resistance to erythromycin[43], vancomycin[44], and tetracycline[45], respectively.

## Highly coupled virus−host links in different habitats

Predicting the cellular hosts of viruses is important for understanding the dynamics of virus−host interactions and potential coevolution mechanisms; hence, both in situ and ex situ hosts were identified to maximize host assignment[21]. The obtained ex situ virus−host links covered 122 vOTUs, accounting for 31% of the genomes (Supplementary Data 5, Fig. S6a). Only bacteria were considered when predicting

ex situ hosts according to genome–spacer matches; the dominant hosts were the bacterial genera *Corynebacterium*, *Barrientosiimonas*, *Micrococcus*, and *Kocuria*, which are members of the class Actinomycetia (Fig. S6b). To systematically elucidate the interactions between viruses and their in situ hosts, an extensive assembly of microbial genomes from the same dataset was performed, which resulted in 58% of the viral genomes being linked to a set of 860 representative metagenome-assembled genomes (rMAGs) (Supplementary Data 6, Fig. S6a). Two other methods—WIsH[46] and tRNA matches—were also used to identify the links between rMAGs and viruses, which resulted in an additional 17% of the viral genomes being matched to in situ hosts (Fig. S6a). Altogether, in situ hosts were predicted for 349 vOTUs or 81% of the viral genomes (Supplementary Data 7), which was ~3-fold higher than the number of vOTUs or the percentage of viral genomes for which ex situ hosts were predicted, indicating that viruses in the BEs have a narrow host range. A network for the in situ virus–host links at the family level was further constructed, and the most frequently predicted hosts belonged to the family Mycobacteriaceae, followed by Dermatophilaceae and Micrococcaceae (Fig. 4a).

To further investigate the potential influences of viruses on microbial ecology in BEs, the viral infection dynamics of specific host lineages across habitats were assessed based on the lineage-specific virus–host abundance ratios at the family level of the hosts (Fig. 4b). Among the different lineages, a range of virus–host abundance ratios were observed, with the relative abundances of the viruses often being below those of the hosts, except for the bacterial family Parvularculaceae (Fig. 4b). Most lineage-specific virus–host abundance relationships (31 of 42) differed significantly between the habitats (Supplementary Data 8). For example, significant correlations with high Pearson's coefficients ($\geq$0.75) were found between virus and host abundances for the bacterial families Caulobacteraceae, Dermabacteraceae, and Dermatophilaceae on residential surfaces and for the family Chroococcidiopsidaceae on pier surfaces (Fig. 4b, Supplementary Data 8). However, the taxonomic distribution of hosts and viruses varied significantly (ANOVA, $p < 0.01$) across habitats in the piers, whereas it was relatively homogenous across habitats in the residences (Fig. 4b). Interestingly, lytic-cycle-related proteins were more prevalent in viruses with higher virus–host abundance ratios (Fig. 4b), suggesting that a potential increase in lytic viral infection reduces host growth and abundance. Conversely, more viruses with lysogenic cycles were linked to hosts that were dominant (e.g., Micrococcaceae and Dermatophilaceae) in most of the habitats (Supplementary Data 2, Supplementary Data 7), supporting the Piggyback-the-Winner hypothesis[22]. Notably, a smaller proportion of viruses with lysogenic cycles was observed in many of the habitats that experienced harsh environmental conditions, such as air in subways and floors and handrails in piers (Fig. S7).

## Evidence of CRISPR-Acr interactions in the BE viromes

Highly coupled virus–host links can lead to CRISPR-Acr interactions in prokaryotes[47]. To investigate these interactions, 2,478 CRISPR spacers were extracted from 25% of the rMAGs and found to be prevalent in members of the bacterial families Micrococcacea, Mycobacteriaceae, and Deinococcaceae (Supplementary Data 7). However, only 2% of the spacers were linked to the viral genomes in the same dataset. In particular, a complete CRISPR/Cas system was identified in rMAGs with a provirus integration (Fig. 5a), suggesting that Acr proteins were prevalent in the viruses to parry the CRISPR/Cas defense. To validate this, the Cas-encoding genes in all of the rMAGs were first identified: 34 rMAGs harbored types I and III CRISPRs associated with the Cas1, Cas2, and Cas10 systems, with type I being the dominant CRISPR type (Supplementary Data 9). Next, PaCRISPR[48] was used to predict Acr proteins from the 99,084 viral proteins and 6,283 putative proteins that were found. After filtering out the Acr proteins based on their adjacent helix-turn-helix (HTH) domain-containing proteins and

alignment against the Protein Data Bank and Conserved Domains Database[49], 162 protein families were identified as potential Acr proteins, and most of them were found to be carried by Caudoviricetes viruses (Supplementary Data 10). Overall, several candidate Acr types were identified (i.e., AcrIB, AcrIF, and AcrIIA), with type I Acr proteins forming the largest Acr cluster due to being prevalent across different habitats (except air) (Fig. S8). The AcrIA cluster was also consistent with the major type I Cas cluster found in the MAGs (Supplementary Data 9).

Based on the spacer-matched virus–host link, we found evidence of a type I Acr protein involved in type I CRISPR/Cas system inhibition. A Siphoviridae virus (SL336563_c_18_full) obtained from an occupant's palm was found to carry an Acr0001-encoding gene (Fig. 5b), which, according to a homology search against the Acr curated database[48] and a phylogenetic tree (Fig. 5c), can be used to evade type I CRISPR/Cas immunity. This result is consistent with the type I–E CRISPR/Cas systems found in doorknob-borne hosts from the same residence. The evolution of CRISPR resistance can cause rapid extinction of phages, especially when the CRISPR/Cas systems feature a high diversity of spacers[25]. Interestingly, no integrated phage was found in an rMAG (SL336752_bin.1_c_05) in which only one contig harbored four CRISPR loci with 86 unique CRISPR spacers.

Nine of the 162 predicted Acr proteins were considered novel, i.e., they did not match a known reference with a BLAST-based hit E-value < 0.001 (Supplementary Data 10). A phylogenetic tree was constructed to further determine the uniqueness of all of the predicted Acr proteins, which were found to be broadly distributed in different sub-types (Fig. 5c). Computational modeling of all of the predicted Acr proteins revealed diverse structures (48 were considered as having high confidence [pIDDT > 80]) (Fig. S9). Comparison of the high-confidence structures of the four novel predicted Acr proteins with their closest references revealed differences, which may be responsible for the variations in their functions (Fig. 5d). Several predicted Acr proteins were located in complete circular vOTUs (with lysogenic and lytic cycles) of unknown families (Fig. S10a-b), suggesting that these proteins play roles in the evolution of poorly characterized Caudoviricetes viruses. Some Acr proteins were located between the integrase and terminase subunits in viruses with lysogenic cycles (Fig. S10a), while others were close to the terminase subunits in viruses that make lytic cycles (Fig. S10b), indicating that the Acr-encoding genes are expressed not only upon initial entry and during lysogeny but also upon transition to the lytic cycle to prevent the cleavage of progeny phage genomes by CRISPR/Cas systems, as previously demonstrated in *Listeria* phages[50]. Additionally, a set of lytic genes, including those encoding endolysins, the Rz lysis protein, holin, and holin–antiholin, were carried by specific complete circular vOTUs with lysogenic cycles (Fig. S10a). Conversely, one complete circular vOTU (SL336690_c_82_full) that makes lytic cycles still harbored an integrase (Fig. S10b), suggesting that this virus undergoes a transition from a lysogenic to a lytic cycle if the environmental condition changes[51].

## The AMGs of viruses are linked to specific hosts

Studies have reported that viruses in many ecosystems possess AMGs that participate in hosts' metabolism to facilitate their adaptation to the environment[3,4]. The AMGs in the BE viromes were investigated, and 468 putative AMGs were recovered, including 86 with unknown functions (Supplementary Data 11). Most of the AMGs were found to play putative roles in the metabolism of cofactors and vitamins (24%), carbohydrate metabolism (22%), and amino acid metabolism (18%). Notably, the AMGs carried by viruses in indoor habitats were clearly distinct from those in outdoor habitats (Fig. 6a). The essential enzyme dUTPase, which is involved in regulating the cellular levels of dTTP/dUTP and crucial for the fidelity of DNA repair and recombination[52], was found to be encoded in some viruses of the BEs (Fig. 6a, Fig. S10a,

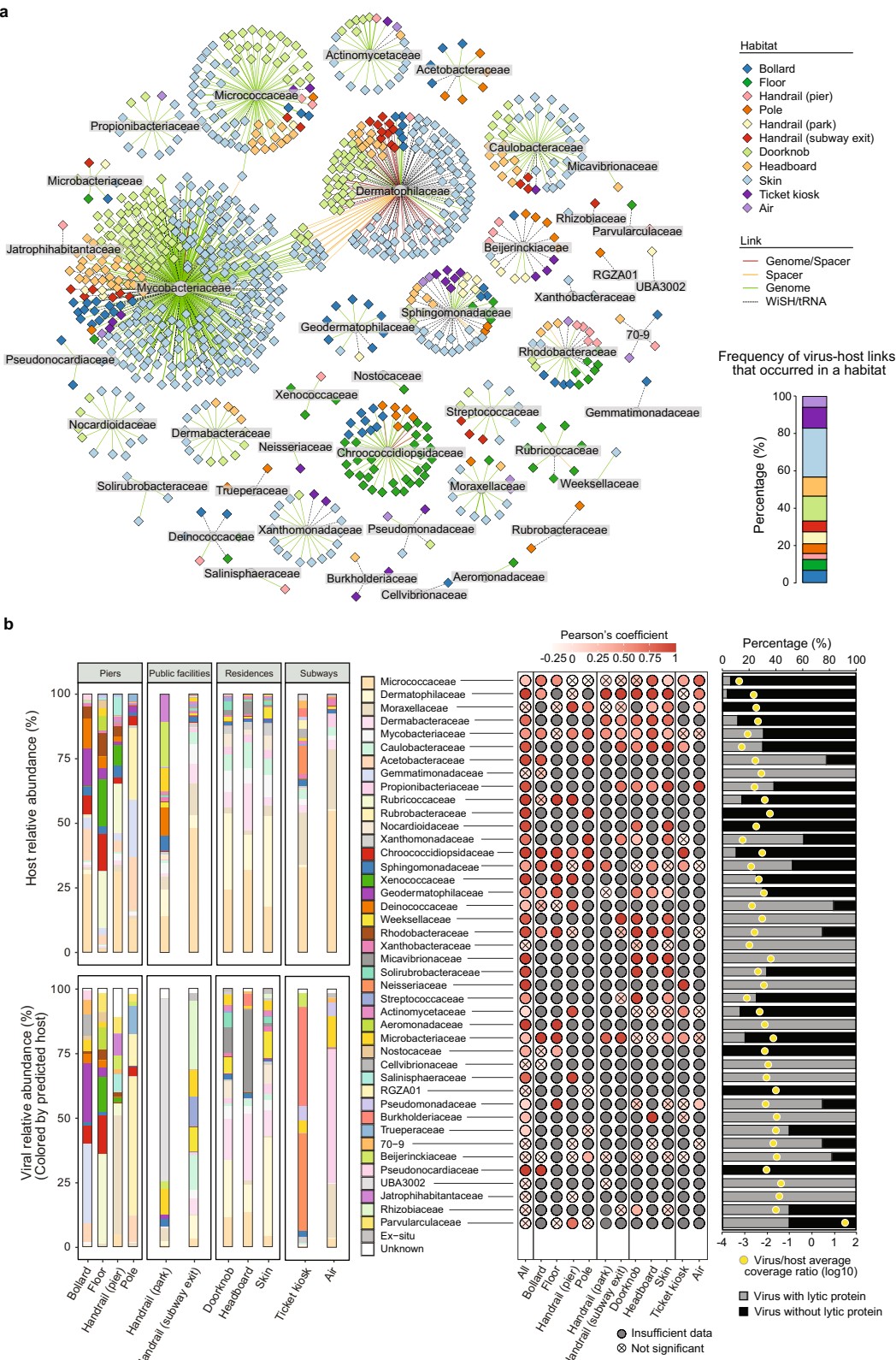

**Fig. 4 | Virus–host links in the different habitats of built environments (BEs). a** A network diagram illustrating viruses and their predicted bacterial hosts at the family level. The circles and diamonds indicate the predicted bacterial hosts and viruses, respectively, and the edges are colored according to the prediction methods. The frequency of virus–host links occurring in a habitat is shown on the right. **b** The relative abundances of hosts and viruses (grouped by predicted host taxonomy) based on the read mapping of the 614 bulk metagenomes from BEs (left panel). Taxonomy is ordered by the average host coverage across all of the habitats. Pearson's correlation of the pairs of relative virus–host abundances in each sample (middle panel). Significantly correlated relative abundances ($p < 0.05$) are colored according to Pearson's coefficient (middle panel). The average virus–host abundance ratios across the 614 samples are shown (yellow circle), and the percentages of viruses with and without lytic proteins (bar plot) are shown (right panel). The exact $p$-values are shown in Supplementary Data 8.

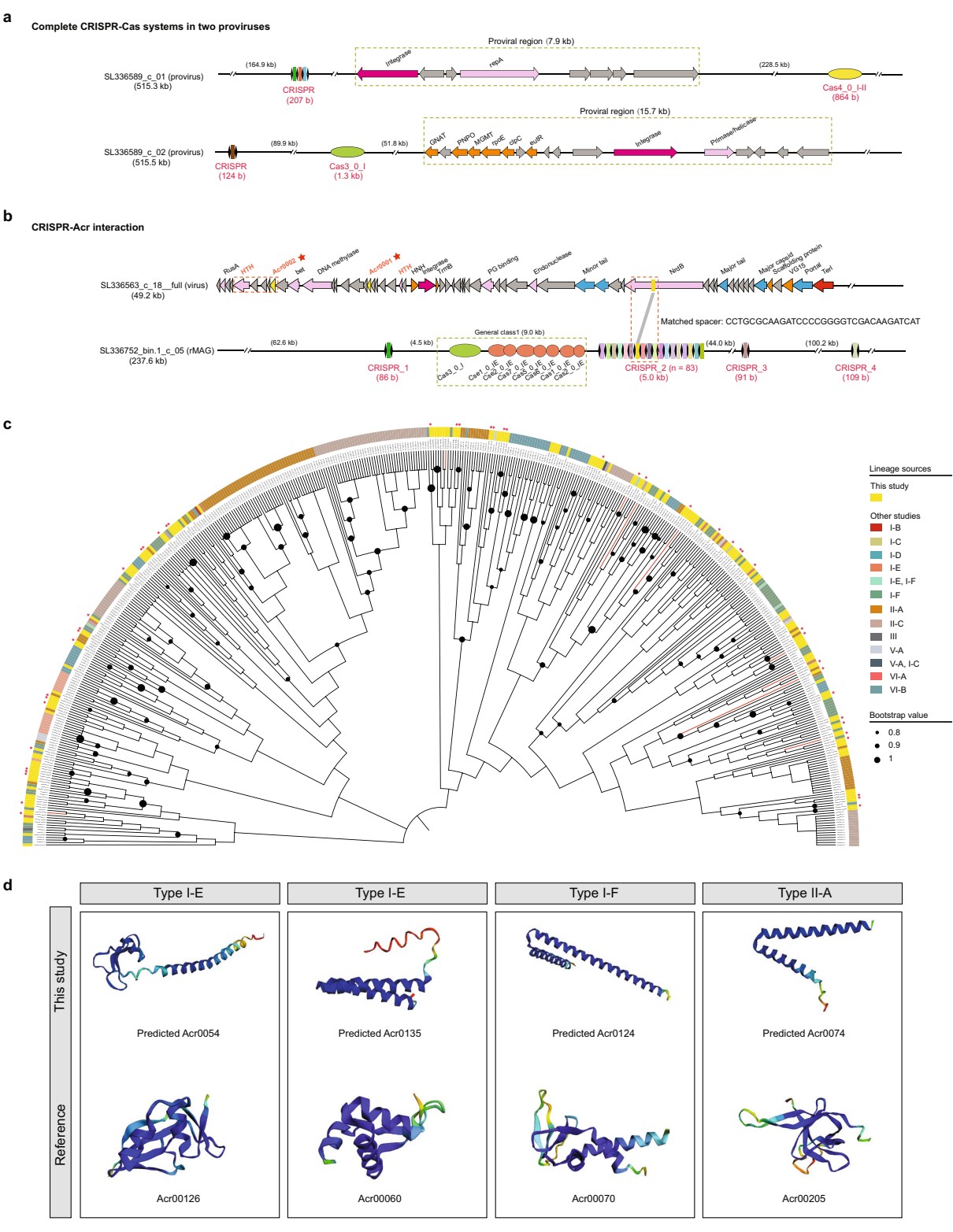

**Fig. 5 | The clustered regularly interspaced short palindromic repeat (CRISPR)–CRISPR-associated protein (Cas) systems and the predicted anti-CRISPR (Acr) proteins in the built environment viromes. a** The complete CRISPR/Cas systems identified in two proviruses. The CRISPR locus is highlighted in red. **b** The potential Acr mechanism found in a virus that could be linked to a host using spacer matching is highlighted by a red dashed box. The Acr protein is indicated by a star, and the CRISPR locus is highlighted in red. **c** A maximum- likelihood phylogenetic tree of the 162 Acr proteins identified in this study and the 339 Acr proteins obtained from the curated PaCRISPR database[48]. The branches of the nine novel predicted Acr proteins are shown in red. The red stars on the outer ring denote the predicted Acr proteins with high-confidence structures (plDDT > 80) based on the AlphaFold2 tool. **d** Comparison of the high-confidence structures (plDDT > 80) of the four novel predicted Acr proteins with their closest reference.

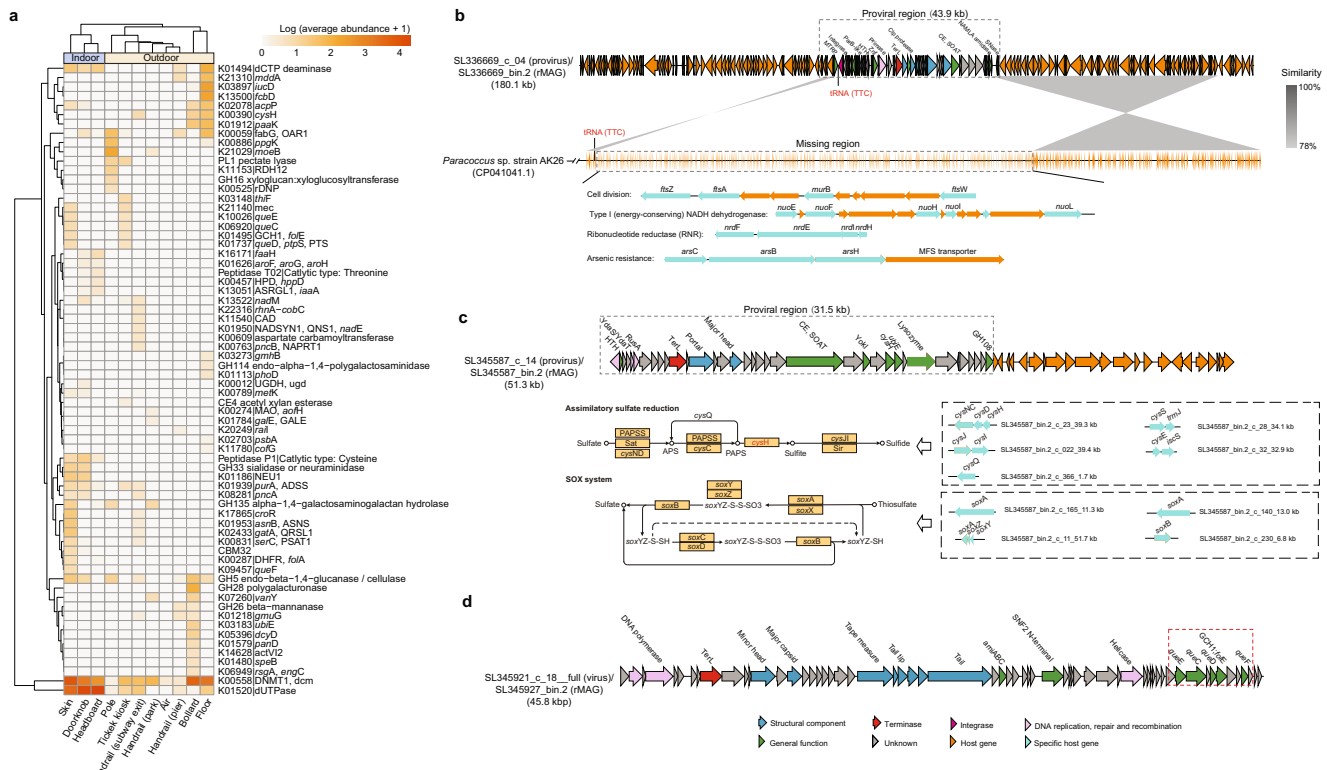

**Fig. 6 | Characterization of auxiliary metabolic genes (AMGs) in the built environment (BE) viromes. a** A heatmap of the average abundance of each AMG in the BE habitats. The average abundance was calculated based on the average abundance of the viruses harboring the corresponding AMGs in each habitat. **b** A schematic that illustrates the insertion of genes by a virus disrupting the host genome. The region of genes in the host that was missing based on comparative genomic analysis with a closely related strain is highlighted by a gray dashed box. **c** A schematic that illustrates the insertion of two AMGs controlling a specific step of sulfate reduction by a virus into a host. The proviral region is highlighted by a gray dashed box, and the AMGs are highlighted by a red dashed box. The host genes (black dashed box) and pathways involved in sulfur metabolism are shown. **d** A schematic that illustrates the insertion of five AMGs encoding the entire preQ$_0$ biosynthesis pathway by a virus into a host. The AMGs are highlighted by a red dashed box.

b). However, it is unclear why viruses encode an enzyme that is prevalent in eukaryotes and prokaryotes[52].

Upon cell infection, some viruses integrate their genomes into the host chromosome, disrupting the host genes[19]. Specifically, a provirus (SL336669_c_04) was found in a host whose genome was highly similar (83.5% ± 5.7%) to the reference *Paracoccus marcusii* (CP041041.1), except for the region with the viral insertion. The genes inserted by the provirus encode cold shock proteins, ABC transporters for sugar and carbohydrate, and proteins that function in arsenic resistance and energy conversion[53] (Fig. 6b). The viral insertion had also disrupted the *fts*AWZ genes encoding cell division proteins and the *nrd*EFHI genes encoding a ribonucleotide reductase that provides deoxyribonucleotides for DNA synthesis and repair in the host genome. A comparison of the genomes between the provirus identified here and a known *P. marcusii* phage[54] revealed no similarity.

In addition to genome insertion, there was evidence of viral insertion of AMGs into the genomes of two hosts that possibly altered the hosts' metabolism and enhanced their adaptability to a habitat. On the surface of a bollard on a pier, a provirus (SL345587_c_14) was found to have introduced two AMGs (*cys*H and *ubi*E) into the genome of a host (SL345587_bin.2) that belonged to the family Acetobacteraceae (Fig. 6c). *Cys*H encodes a 3′-phospho-adenylylsulfate reductase that generates sulfite by transformation of sulfate, which is part of the hydrogen sulfide biosynthesis pathway in sulfur metabolism[55]. Other genes responsible for sulfur metabolism were also identified in the rMAG, including *cys*NCDH that is involved in regulating 3′-phosphoadenosine-5′-phosphosulfate (PAPS) formation, *cys*JI that encodes sulfite reductase in the assimilatory sulfate reduction pathway, *cys*Q that

generates an adenosine 5′-phosphosulfate reductase from PAPS, and other *sox*ABZY genes involved in the sulfur oxidation system (Fig. 6c). In addition to synthesizing sulfite, *cys*H can also control the pool of cellular PAPS, which is toxic if allowed to accumulate[56]. Another indoor-doorknob-borne host belonging to Mycobacteriaceae (SL345927_bin.2) was linked to a viral genome (SL345921_c_18_full) derived from the right palm. The AMGs (*que*CDEF and *fol*E) that encode enzymes involved in the entire 7-cyano-7-deazaguanine (preQ$_0$) biosynthesis pathway were found to have integrated into the host genome (Fig. 6d). The genes encoding preQ$_0$ biosynthesis were not found in the available Mycobacteriaceae genomes (from NCBI: txid1762). Taken together, these findings suggest that the metabolism of hosts in BEs can be influenced by viral infection, which may alter their adaptability to a specific habitat.

## Discussion

Viruses, despite being as numerous as bacteria in BEs[8], have received far less research attention in the literature. A recent global study of surface microbiomes in urban environments showed that it is feasible to recover diverse viruses from bulk metagenomic samples from BEs[15]. In this study, we analyzed in detail the viromes derived from 11 habitats across four types of BEs in HK.

Several vOTUs were identified in each BE habitat, but their diversity was significantly lower than that of vOTUs previously identified in global marine water[36]. The material and type of a given surface is likely to be the key drivers of virome diversity, with concrete, wood, and skin surfaces harboring a higher diversity than metal and plastic surfaces, as indicated by our findings. Even on surfaces made of the

same metal, contact frequency plays a major role in controlling virome diversity, as indicated by a frequently touched indoor doorknob exhibiting a higher virome diversity than the sparingly touched outdoor handrails. Unlike marine environments with abundant resources[20], most habitats in BEs have poor nutrient supply and uncontrolled and harsh environmental conditions comprising intense ultraviolet light and fluctuating temperature and humidity. These unfavorable conditions, together with the properties of surface materials, can drive taxonomic variations and functional shifts in the bacterial microbiome[9]. Ecological evidence also supports that environmental filtering can control the bacterial diversity in indoor environments[57], which may in turn affect the diversity of phages that have a narrow host range[58]. Consistent with the differences in diversity and structure between bacterial communities in air and on surfaces[59], the airborne and surface-borne viromes in the present study were distinct. The use of advanced ventilation and filtration systems may explain the lower virome diversity in subway air than in air from venues using natural ventilation[12]. Viruses belonging to the class Caudoviricetes, many of which remain unclassified at the family or genus level, were ubiquitous and dominant across most habitats except air in the HK BEs, consistent with the findings for other ecosystems[30].

Accumulating evidence suggests that viruses can survive in harsh environments and aid the survival of their bacterial hosts[60]. In HK BEs, highly coupled interactions between phages and bacterial hosts were observed, and the proportion of host-linked viruses was more than 2-fold higher than that in soil[17] and the human gut[30], both of which have a rich nutrient supply. After invading a host, a phage can apply different strategies to drive host adaptation[18]. One mechanism is the insertion of large DNA fragments into highly conserved insertion sites of the bacterial chromosome by phages with lysogenic cycles[61], which, while having deleterious effects on host fitness, can also generate genetic variations for evolutionary innovation to aid host adaptation to new environments. Another mechanism is the expression of phage-encoded AMGs after insertion into the host genome[3]. The inserted AMGs vary depending on the environmental conditions to which the host is exposed[21], and this phenomenon was also observed in different BE habitats. On frequently touched indoor surfaces, the AMGs that encode dUTPase were prevalent, likely aiding viral replication to allow them to persist effectively in their hosts[62]. On a pole in a pier that frequently receives splashes of marine water, the AMG cysH, which participates in sulfur metabolism, was integrated into a host by a phage with lysogenic cycles. Interestingly, this gene has also been identified in viral sequences obtained from oxygen-deficient water columns[63] and a deep freshwater lake[64]. CysH potentially assists the host to overcome a reaction bottleneck in sulfur metabolism by regulating a specific step of the metabolic pathway. In contrast to the insertion of a single AMG, genes involved in the regulation of the entire preQ$_0$ biosynthesis pathway for the formation of a diverse class of nucleoside analogs possessing antibiotic, antineoplastic, or antiviral activities[65] were found in a bacterial host on human skin and a ticket kiosk. The synthesized compounds in this host may aid host adaptation to frequently touched surfaces. Several AMGs of unknown function were also found in the BEs, and they should be further investigated experimentally to determine their roles in virus–host interactions.

The viral life strategy of switching between lysogenic and lytic cycles is a key driver of virus–host evolution[51]. Generally, the abundance of viruses with lysogenic cycles varies according to environmental conditions, with the abundance being higher under conditions of lower bacterial density and nutrient levels[66]. While both indoor and outdoor habitats in BEs are expected to favor viruses with lysogenic cycles, a relatively lower proportion of these viruses was observed in outdoor habitats. Generally, outdoor habitats are more vulnerable to external stresses than internal habitats, resulting in the expression of lytic genes and transition from a lysogenic to lytic cycle under stressful conditions (i.e., DNA damage)[51]. Similar to the marine environment[3],

the release of organic matter via viral lysis may play a role in nutrient and resource cycling for the members of microbial communities in oligotrophic BEs. However, future studies are required to determine the ecological model viruses use for establishing themselves in the BEs.

Phage-associated ARGs are a concern due to the possibility of their wide dissemination through hijacking hosts' genome replication machinery[67]. Compared with bacteria-associated ARGs, phage-associated ARGs pose a more serious threat, as their dissemination routes are challenging to track and predict[67]. Although phage-mediated ARG transduction is rare and the ARGs may not be functional[37,68], antibiotic resistance conferred by functional beta-lactamase-encoding genes has been identified in freshwater viromes and validated experimentally[69]. In the BEs in our study, most putative ARGs were found in viruses inhabiting human skin or frequently touched indoor surfaces. These ARG-carrying viruses may infect bacterial hosts, and subsequently, the putative ARGs may be horizontally transferred between bacterial species[70]. In addition to ARGs that confer resistance to beta-lactam antibiotics, several AGRs that confer resistance to macrolides, vancomycin, and tetracycline were also identified in the viruses from the BEs. Thus, the role played by viruses in the development of antibiotic resistance in bacteria, especially those present on frequently touched surfaces in BEs, is crucial and warrants further investigation.

The frequent battle for survival between bacteria and phages has led to the evolution of defense systems in many bacteria[19]. Particularly, how phages inactivate the CRISPR/Cas systems in bacteria via Acr proteins has received growing attention because CRISPR/Cas systems are the only adaptive immune system identified in prokaryotes to date[50]. In this study, several Acr proteins were predicted in phages found in the BEs. Given their function in counter-defense, phage-encoded Acr proteins evolve rapidly and show limited sequence similarity to experimentally characterized Acr proteins[71], making inference of the type of Acr proteins challenging. Consequently, most of the Acr proteins identified in this study are not targeted by the CRISPR/Cas systems of bacterial hosts. However, Acr proteins may not be as effective as once believed because CRISPR system resistance also evolves rapidly and may thus cause rapid extinction of phages, especially if the CRISPR/Cas systems develop a high diversity of spacers, which is difficult for phages to overcome by point mutations[25]. This phenomenon is consistent with our results that no proviruses were found in an rMAG that contained four CRISPR loci with 86 spacers.

While this study has shed light on viruses in BEs, it has a few limitations. First, potential biases could occur in our virome analysis workflow. Owing to the low biomass in the BE samples and the potential biases associated with the DNA extraction and purification methods, the diversity of viruses and their hosts could be underestimated. Furthermore, the lack of appropriate references in the bioinformatics tool databases could lead to poorly characterized sequences[72]. Second, the results derived from metagenomic sequencing did not differentiate whether the identified viruses and their associated genes (e.g., ARGs and Acr proteins) were functional or defective. Future experimental investigations will be required to identify the functions of the novel Acr proteins. Third, RNA viruses were not considered; thus, the full virome diversity in the BEs could not be examined. Fourth, while the applied contig-based tools[27,28,30] enabled us to uncover many novel, previously uncultivated viruses, assembly with other algorithms may reveal other viruses, and a viral binning method may better address fragmented multi-contig viral assemblies to enable more precise clustering of both viral and bacterial populations and direct investigation of virus–host interactions[73]. Lastly, future comparisons with other highly-selective, frequently cleaned, and resource-limited environments, such as hospitals and medical wards[74], could further contextualize the influence of physical and chemical properties of surfaces on the abundances of viruses and the virus–host interactions.

In summary, we believe that this study is the first to show the diversity, taxonomy, metabolic functions, and lifestyles of viromes across diverse habitats in BEs. This study highlights the tight coupling between viruses and bacterial hosts in BEs and illustrates the significance of their coevolution through different virus–host interaction mechanisms for host adaption and virus survival in oligotrophic BEs. Many novel viruses from the class Caudoviricetes and Acr proteins were also identified, suggesting that more of these remain to be discovered in BEs. Overall, viruses are important members of BE microbial communities, and a greater understanding of their biology can ultimately facilitate better design of cities and protection of public health.

## Methods

### Sample collection and metagenome sequencing

The metagenomic datasets comprised 738 samples collected from 11 habitats across four types of BEs (namely piers, public facilities, residences, and subways) in Hong Kong. From nine piers with low occupancy, 175 samples[9] were collected from the surfaces of four types of habitats: bollards ($n = 40$), floors ($n = 45$), handrails ($n = 45$), and poles ($n = 45$). From eight public facilities (i.e., four parks and four subway exits) with medium occupancy, 134 samples[75] were collected from the surfaces of two types of habitats: park handrails ($n = 69$) and subway exit handrails ($n = 65$). From four residences each with a single occupant, 268 samples[75] were collected from the surfaces of three types of habitats: doorknobs ($n = 66$), headboards ($n = 68$), and occupants' skin (i.e., left and right palms and forearms; $n = 134$). Written informed consent to collect skin samples were obtained from the occupants and the study was approved by the City University of Hong Kong Human Subjects Ethics Sub-Committee (ref: H001553). From 84 subway stations with high occupancy, 161 samples[76] were collected from two types of habitats: surfaces of ticket kiosks ($n = 81$) and air above the platforms ($n = 80$). The same sampling method was used to collect all of the surface[75] and air[76] samples. The materials of the surfaces were either metal, plastic, or concrete. Detailed information about the samples is presented in Supplementary Data 1.

The genomic DNAs of all of the surface and air samples were extracted using the corresponding methods[75,76]. In brief, genomic DNA was extracted from swabs collected from sampled surfaces using the ZymoBIOMICS 96 MagBead DNA kit (Zymo Research, CA, USA), following the manufacturer's instructions. The cells were lysed using a chemical solution and mechanical bead beating technology. Genomic DNA was extracted from air sampling filters using a customized protocol. Cell lysis was performed using NucliSENS Lysis Buffer (BioMérieux, Marcy-l'Étoile, France) and a multi-enzyme cocktail (MetaPolyzyme, Sigma-Aldrich, MO, USA), followed by mechanical bead beating. A ZymoBIOMICS synthetic microbial community standard (Zymo Research Corporation, CA, USA) was used as the positive control in the extraction process and sequenced in tandem with the surface and air samples. Sequencing of all of the genomic DNA samples was performed on an Illumina HiSeq X Ten System (Illumina Inc., San Diego, CA) at HudsonAlpha Genome Center (Huntsville, Alabama)[15]. Quality control and assembly of reads into contigs were performed as previously described[9]. Briefly, adapters were removed from the raw sequences using AdapterRemoval (v.2.2.2)[77]. Quality filtering and trimming were performed using KneadData (v.0.7.6) with default parameters and the human genome hg38 as the reference to remove human sequences. The positive controls yielded the expected sequencing results. Reads in a sample that could be mapped to contigs in the negative controls were removed using an in-house script, and any unpaired reads were further removed from the paired-end fastq files using fastq-pair (v.1.0; https://github.com/linsalrob/fastq-pair). All of the contaminating species identified by decontam (v.1.12; https://github.com/benjjneb/decontam) based on the default threshold were removed. After quality control, ~5.0 million paired-end clean reads per

sample were retained and assembled into ~4.5 million contigs using MetaWRAP (v.1.2.1)[78].

### Recovery of viral contigs

The performance of two viral detection tools was first evaluated on a mock dataset containing 2000 randomly selected genome fragments (1, 3, 5, 10, and 20 kb) of bona fide viruses from the RefSeq viral database (v.209). The average viral recall of the genomic fragments of different lengths was calculated using VirSorter2[27] (v.2.2.4; all categories) with the default cutoff and DeepVirFinder (v.1.0)[28] with different cut-offs (0.5, 0.7, 0.8, and 0.9) (Fig. S1c). Based on the mock dataset results, VirSorter2 with the default cutoff (all categories) and Deep-VirFinder with a predicated score ≥0.5 and a $p$-value < 0.05, which resulted in the highest viral recall for each fragment length, were used in tandem to process the 4.5 million contigs assembled from 738 metagenome samples (Fig. S1b). In total, 594,851 unique putative viral contigs were obtained from the two tools. CheckV (v.0.8.1; database v.1.0)[79] was used to assess the quality of the putative viral contigs, and 1174 viral contigs that included complete, high-quality (>90% completeness), and medium-quality (50–90% completeness) genomes were retained[30] (Supplementary Data 2). For genomes that contained predicted proviruses, only the proviral regions were retained. PHA-STER (https://phaster.ca/)[80] and VIBRANT (v.1.2.1)[81] were separately applied to identify proviral sequences according to the following two criteria, as previously proposed[21]: (i) the viral contigs were from contigs with non-viral (host) flanking sequences or (ii) the viral contigs harbored lysogenic marker proteins (i.e., integrase and serine recombinase). In total, 332 unique proviruses were identified using the two aforementioned tools and CheckV, and only those (127) that were integrated into a bacterial genome had high confidence (the prophage region's total score was >90 in PHASTER) and were included in the downstream analysis.

### Viral genome clustering and taxonomic assignment of viral operational taxonomic units (vOTUs)

All viral genomes with >50% completeness were clustered into species-level vOTUs on the basis of 95% average nucleotide identity (ANI) of >85% alignment fraction relative to the shorter sequence based on centroid-based clustering[30]. Genus- and family-level vOTUs were generated using a combination of gene sharing and amino acid identity (AAI) based on Markov clustering[82] as described previously[30]. Briefly, viral genomes with <20% AAI or <10% gene sharing and an inflation factor of 1.2 were clustered into family-level vOTUs, while those with <50% AAI or <20% gene sharing and an inflation factor of 2.0 were clustered into genus-level vOTUs.

The open reading frames (ORFs) in the 471 vOTUs were predicted by Prodigal (v.2.6.3)[83] using the parameter "-p meta." The protein-coding gene sequences of the 471 vOTUs were assigned family-level taxonomy using the majority-rule approach as previously described[30]. Specifically, the taxonomy based on the International Committee on Taxonomy of Viruses (ICTV) of the top IMG/VR (v4) database[29] hit using DIAMOND (v.0.9.32; options: –query-cover 50 –subject-cover 50 –E-value 1e$^{-5}$ –max-target-seqs 1000) was transferred to each protein. In cases where the taxonomy of the top hit was missing, the next hit was adopted if its bit-score was within 25% of the top hit. Each vOTU was then assigned to the lowest taxonomic rank of >70% of the annotated proteins. At the family and genus ranks, a genome must have a minimum of two annotated proteins with >30% average AAI or three annotated proteins with >40% average AAI, respectively, aligned to a reference genome from the IMG/VR database[30].

### Estimation of viral coverage

The clean reads from all of the 738 metagenomes were mapped to the viral genomes using Bowtie2 (v.2.4.4) with the default parameters. A BamM (v.1.7.3; http://ecogenomics.github.io/BamM/) "filter" function

was used to screen the reads that were mapped to the genomes to remove low-quality mappings, and reads that aligned over ≥90% of their length at ≥95% ANI were retained. The viral genomes with ≥70% of their length covered by the reads were selected using a Python script in Read2RefMapper (v.1.1.0), and the average per-base-pair coverage of each contig in each sample was generated using the BamM "parse" function with the parameter "tpmean" to remove the highest and lowest 10% coverage regions. Based on the mapping analysis, viral genomes were detected in 614 metagenomes. The average relative abundance of the 1174 viral genomes across the 614 metagenomes was calculated by dividing the average coverage of a viral genome by the total number of clean reads across all samples and then multiplying by the average number of all clean reads across the 614 metagenomes to bring the total number of reads for each sample up or down to the average[17].

Three publicly available virome datasets, namely skin metagenomes primarily from subjects in North America (SMGC dataset)[35], ocean metagenomes from temperate and tropical epipelagic and mesopelagic ocean (GOV dataset)[36], and surface metagenomes from subway stations of international cities except Hong Kong (MetaSUB dataset)[15], were analyzed together with the virome dataset generated in this study. The viral genomes retrieved from the public databases were assessed using CheckV, and only medium- and high-quality viruses were retained (Fig. S6a). The mapping of reads to our dataset and the three publicly available virome datasets was performed using the "bowtie2-build" function by first creating four indexes using only species-level vOTUs from all of the virome datasets. The clean reads were then aligned to each genome index using Bowtie2 with the option "–very-sensitive -k 20", and alignments with mapping ANI < 95% were discarded.

## Alpha and beta diversity analyses

Alpha diversity indexes, namely richness, Shannon's H, and Pielou's evenness, were calculated using the R package vegan (v.2.5.7)[84]. Seventy metagenomes with singleton vOTUs were excluded from the analysis. The Mann–Whitney U test was performed to test the statistical significance of the difference between two groups, while analysis of variance was applied to determine the differences between two or more groups. For beta diversity analysis, a principal coordinate analysis based on the Bray–Curtis dissimilarity matrix was performed using the "vegdist" function of the R package vegan. Pairwise analysis of similarity and permutational multivariate analysis of variance were performed to test the significance of dissimilarity between groups by using the "anosim" and "adonis" functions of the R package vegan, respectively.

## Construction of phylogenomic trees for Caudoviricetes

A maximum-likelihood phylogenetic tree comprising Caudoviricetes vOTUs from this study and the three publicly available virome datasets (SMGC, GOV, and MetaSUB) was constructed as described previously[30,85]. Briefly, 77 curated Caudoviricetes markers were searched against the protein-coding gene sequences of the vOTUs using a profile hidden Markov model (HMM). The dataset from this study contained 444 vOTUs with lengths ≥5 kb and 37,775 protein-coding gene sequences, while the three public datasets contained 2389 vOTUs with lengths ≥5 kb and 168,256 viral protein-coding gene sequences. The top HMM hits were individually aligned to the profile HMMs of the 77 markers, as previously recommended, using the "hmmsearch" function, which retained 40 and 58 markers from this study and the four datasets combined, respectively. The alignments of individual markers were then trimmed using trimAl (v.1.4)[86] to retain positions with <50% gaps, and gaps were filled where necessary using an in-house Python script. Only genomes with >5% amino acid representation in the total alignment length were retained. A concatenated protein phylogenetic tree was inferred from the multiple sequence alignment using FastTreeMP (v.2.1.11) with the auto model[87]. The

tree was midpoint-rooted and visualized using iToL (v.6; https://itol.embl.de/).

## Functional annotation of viral contigs

The ORFs in the viral genomes were annotated against several protein family databases, including KEGG[39], Pfam[38], TIGRFAM[88], VOGDB (http://vogdb.org), and the Earth's Virome database[89], using the profile HMM search method performed using the hmmsearch utility in HMMER (v.3.1b2) with the default parameters. The annotation of the top-scoring alignment (bit-score ≥ 60 and an E-value ≤ 1e−5) among the databases was assigned to each ORF.

## Annotation of antibiotic resistance genes (ARGs)

The ARGs in the viral genomes were annotated using the Resistance Gene Identifier tool (v.5.2.0)[42] with the option "–low_quality," which applied the best identity of >60% to the reference sequences in the CARD database (v.3.0.9)[90], and using the NCBI AMRFinderPlus (v.3.8.4) tool[41], with the default options of 60% coverage and 80% identity, to the reference sequences in the Resfams database (v.1.2)[40]. The search was performed using the hmmsearch utility in the HMMER tool (v.3.1b2), with an E-value ≤ 1e−5 and a gathering threshold score ≥40. The Resfams annotation with the best score was adopted when an ARG received different annotations from the databases.

## Annotation of auxiliary metabolic genes (AMGs)

AMGs were annotated based on the viral mode of DRAM (v.1.4.0)[91], which uses the output produced by VirSorter2. The 1174 identified viral contigs were reprocessed by VirSorter2 (--prep-for-dramv) to produce the "VIRSorter affi-contigs.tab" file and then annotated with the default databases in DRAM. This process eliminated 573 viral genomes with low viral scores according to VirSorter2. Putative AMGs in the remaining 601 viral genomes were identified based on a high auxiliary score of 1 or 2. The gene descriptions adopted were based on the distilled annotation of DRAM-v with the default parameters. To supplement the AMG annotations from DRAM, the viral genomes were also annotated using VIBRANT with the default parameters, and annotations not found in DRAM were retained.

## Prediction of anti-CRISPR (Acr) proteins

Candidate Acr proteins were first predicted using PaCRISPR[48] with a default cut-off threshold, and further filtering of the candidate Acr proteins was performed as previously described based on HTH domain-containing proteins that must be located within five genes upstream or downstream[92]. The filtered candidate Acr proteins were subsequently clustered using MMseq2 (v.13.45111)[93] with the parameters "--min-seq-id 0.5 -c 0.7 --cov-mode 2 --cluster-mode 0". The representative Acr proteins that did not produce a hit with an HHpred probability ≥0.9 to any Protein Data Bank and Conserved Domains Database sequence were regarded as predicted Acr proteins and retained for downstream analysis[49]. The types of predicted representative Acr proteins were estimated using a PSI-BLAST search with the default parameters against the Acr curated database PaCRISPR[48]. Multiple sequence alignment between the predicted Acr proteins and the 339 reference Acr proteins obtained from the curated PaCRISPR database was generated using FAMSA (v.1.5.12)[94] and trimmed using trimAl (v.1.4)[86] to retain positions with <50% gaps. A maximum-likelihood phylogenetic tree was constructed with the aligned sequences using FastTreeMP (v.2.1.11) with the auto model and was visualized using iTOL. The structures of the Acr proteins were predicted using the AlphaFold2 tool with the default settings[95,96].

## Metagenome-assembled genomes (MAGs) and coverage estimation

MAGs (Supplementary Data 5) were reconstructed from all of the sampled metagenomes as described previously[9]. Briefly, the clean

reads of each sample were assembled into contigs, and those with lengths >1000 bp were binned into MAGs using MetaWRAP (v.1.2.1)[78]. The resulting MAGs were further refined using the "bin_refinement" function of MetaWRAP[78] and dereplicated using the "dRep dereplicate" function of dRep (v.3.2.2)[97]. In total, 860 bacterial rMAGs with contamination ≤10% and completeness ≥50% were generated. The ORFs in the contigs of rMAGs were predicted using Prokka (v.1.14.6)[98], and the functions were annotated using EggNOG-mapper (v.2.0.1)[99].

To calculate the relative abundance of each rMAG in all of the samples, the clean reads from each metagenome were mapped to each genome using the BamM "make" function. Low-quality read mappings (<75% aligned length of each read and <95% ANI) were removed using the BamM "filter" function, and the coverage of each genome, which was represented as the mean of the number of reads aligned to each position in the contigs after removing the highest and lowest 10% coverage regions, was calculated using the BamM "parse" function in the "tpmean" mode. The relative abundance of each rMAG was then calculated as the average of the coverages of all of its contigs, weighting each contig by its length in base pairs.

### Determination of virus–host links

Both the in situ and ex situ hosts of the viral genomes were identified as previously described[21]. Two host databases were used to establish the virus–host link, in which 203,065 complete bacterial and archaeal genomes representing 146,464 prokaryotic species from RefSeq (downloaded from the NCBI database on November 2021) were used for identifying ex situ hosts, while 860 bacterial rMAGs that were taxonomically annotated using GTDB-Tk (v.1.5.1)[100] were used for identifying in situ hosts. CRISPR spacers were extracted from the two host databases using a custom Python script, and CRISPR-associated protein (Cas)-encoding genes were detected using CRISPRCasFinder (v.4.2.20)[101].

Microbial hosts for the 1174 viral genomes were predicted using a combination of bioinformatic methods that included viral exact matches (or close similarity) to (i) host CRISPR spacers, (ii) integrated viral fragments in host genomes, (iii) host tRNA genes, and (iv) host k-mer signatures. The methods (i) and (ii) were only used for ex situ host prediction. For method (i), BLASTn from blast+ (v.2.9.0)[102] was used to compare CRISPR spacer sequences with the viral genomes, and matches with ≤ 1 mismatch and an E-value ≤ 1e$^{-5}$ were retained. For any CRISPR spacer that had a match in a viral genome, the repeat sequence from the same assembled CRISPR region was compared with all bacterial and archaeal genomes via BLASTn (E-value ≤ 1e$^{-5}$, 100% nucleotide identity, and 95% coverage) to link that CRISPR region (and any viruses harboring spacers in that CRISPR region) to a host. For method (ii), a bit-score threshold of 50 with an E-value ≤ 1e$^{-5}$ and a ≥96% ANI were used for identifying shared genomic regions via BLASTn, and only hits ≥1000 bp were considered, as these criteria have been shown to yield the most confident host prediction[30]. For method (iii), viral and host tRNA genes were predicted by tRNA-scan SE-2.0 using the general and bacterial/archaeal models, respectively, and BLASTn comparison was then performed between the predicted viral and bacterial tRNA genes. The tRNA matches between the viruses and hosts in the dataset were then scored such that an exact match would score higher (high score) than a host tRNA with a single base difference (intermediate score) and a host tRNA with a two-base difference (low score). For method (iv), WIsH (v.1.1)[46] was used for host prediction after masking tRNA sequences on the viral genomes to improve performance. Subsequently, 3,024 viral genomes (downloaded from the NCBI Virus portal in January 2022) whose hosts are invertebrates were used as a decoy database after conservatively excluding viruses known to infect a host genus under prediction. For each viral genome, the WIsH-predicted host with the lowest $p$-value (≤1e$^{-5}$) was retained to be conservative with family-level host assignments. Both CRISPR spacer and genome matches were retained for in situ host assignment and were

given a higher priority than the WIsH and tRNA results. The average lineage-specific virus–host coverage ratios were calculated by dividing the relative abundance of rMAGs by that of the viral genomes. The network of virus–host links was visualized using Cytoscape (v.3.9.0)[103], and subsampling was performed before constructing a network to eliminate potential biases due to an uneven number of metagenomes sampled across the BE habitats.

## Data availability
The raw DNA-sequencing data used in this study have been deposited in the NCBI Sequence Read Archive under BioProject accession numbers PRJNA671748, PRJNA722771, PRJNA561080, and PRJNA881785. The SRA accession number of each sample is indicated in Supplementary Data 1. Publicly available databases used in this study were Resfams (http://www.dantaslab.org/resfams), CARD (https://card.mcmaster.ca/), Pfam (https://www.ebi.ac.uk/interpro/download/Pfam/), IMG/VR (https://genome.jgi.doe.gov/portal/IMG_VR/), VOG (http://vogdb.org), TIGRFAM (https://www.ncbi.nlm.nih.gov/genome/annotation_prok/tigrfams/), KEGG (https://www.genome.jp/kegg/), and NCBI virus/bacteria/refseq (https://www.ncbi.nlm.nih.gov/refseq/). The datasets analyzed including the Earth's Virome (http://portal.nersc.gov/dna/microbial/prokpubs/EarthVirome_DP/), SMGC (https://www.ncbi.nlm.nih.gov/sra/?term=SRP002480), GOV (https://bitbucket.org/MAVERICLab/), and Meta-SUB (https://github.com/dscdorothy/HK_BE_viromic) are available online. The high-confidence structures of the predicted Acr proteins are provided at https://github.com/dscdorothy/HK_BE_viromic.

## Code availability
The supporting code is provided at https://github.com/dscdorothy/HK_BE_viromic.

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

## Acknowledgements

This research was supported by the Hong Kong Research Grants Council Research Impact Fund (R1016-20F) to A.C.K.L., C.K.C., and P.K.H.L. and the General Research Fund (11214721) to P.K.H.L. Support is also from the National Institutes of Health (NIH) grants R01AI151059 and U01DA053941 to C.E.M.

## Author contributions

S.D. performed data analysis, data interpretation, and wrote the manuscript. X.T. and C.E.M. performed data analysis. A.C.K.L. and C.K.C. provided advice on data interpretation. P.K.H.L. conceived the study and supervised the research. All authors have read and approved the final manuscript.

## Competing interests

The authors declare no competing interests.
