## [Peer Review File · Nature Communications]

Reviewer comments, first round –

Reviewer #1 (Remarks to the Author):

Du et al. report on a very large dataset of 738 bulk metagenomes from built environments, that they have scrutinized under the angle of their (pro?)phage content. The number of analyses they have performed is quite large as well. However, in terms of precision of the analyses, and of the vocabulary used throughout to describe the results, some difficulties emerge. At this stage, I cannot recommend publication of this manuscript, for the main reasons listed below.

1. The authors are probably unfamiliar with phage biology, so the vocabulary is confuse throughout. Some examples: a lysogen designates a bacterium hosting a prophage, and not the prophage itself. Phages "are" not lytic, they make lytic cycles. All phages, whether temperate or virulent, make lytic cycles, otherwise they are defective prophages. A prophage needs not to be integrated in the bacterial genome, it can be maintained as a plasmid. So the absence of an integrase or a junction with bacterial DNA is not sufficient to declare a phage virulent.
2. Linked with this vocabulary problem, there is a deeper problem of understanding what is at stake when a bulk metagenome is analysed for its (pro)phage content. The Supplementary Method section is too superficial to inform on very basic points such as how were the samples lysed? Were spiked phage particles introduced and recovered quantitatively? What is studied really? The prophage content, including all defective prophages that are rampant in bacterial genomes? What is the question addressed finally?
3. The authors think they found beta-lactamase genes in their prophage genomes. They seem to be aware that antibiotic resistance genes are rare in phage genomes, and that many studies were flawed because of irrelevant BLAST cut-offs (at least they cite papers about it). I could not find any indication of the cut-off they used, and the arguments in favour of bona fide antibiotic resistance genes, present on functional prophages (rather than defective ones).
4. In general, there are too many studies, and not sufficiently well described. I would rather choose some interesting aspects, and describe them to the exact details of how the analysis was done, so that someone else can repeat them. The Acr detection is one of the original aspects of this work.

Reviewer #2 (Remarks to the Author):

Shicong Du and co-authors have written a very compelling and well interpreted manuscript that appears to be statistically sound and is of very good quality in general. The study provides excellent data further developing our understanding of the diversity of viruses in built environments, sampling from diverse sources, and the viromics analysis uses mostly up to date techniques. This being said, there is very little in the way of discussion on the various biases that can occur at each stage in the viromics process, and this type of information and self reflection is fundamental to effective interpretation of metagenomic virus data. The findings are indeed of significant interest to the viromics community, particularly the discovery of AMR genes in different host-specific built environments, and the CRISPR/anti-CRISPR led virus-host link analysis is particularly striking for reasons such as novelty and ingenuity.

I have the following comments on the manuscript:

Line 27: I think 'Caudovirales' can be a bit triggering for any viral taxonomists, as tailed bacteriophages appear to have more than one origin, I suggest sticking to 'tailed bacteriophages' if this is specifically what you are referring to, which I believe it is.

The introduction focusses a lot on virus-host ecology, and although this makes a fair share of the results, it does not reflect the literature completely fairly. By this, I mean that adding a little more on comparisons between the virome of very different environments and the interpretation (that they are seemingly individualised, and any other interesting points found by the wealth of viromics studies) may improve the introduction to the topic. For a communication, including more on viromics studies and their interpretations of the global virome and it's role may be of more interest and use than explaining the lifestyle of phages, for example. I appreciate that phage lifestyle is

also integral to the introduction of phages as ubiquitous entities that are also prevalent in BEs, I would suggest trying to include both what you have written and suggested extras.

(General comment: please change all 'Bes' to BE to make the acronym consistent throughout)

Line 60: please see up to date inference of the different models of phage success (<https://www.sciencedirect.com/science/article/pii/S1369527422001138>). For example, if one understands these hypotheses then one appreciates that the 'Royal family' model is actually just different kill-the-winner models, and in fact one of these models in the Royal Family can be explained by an arms race dynamic or fluctuating selection dynamic. Also, I would not refer to these as 'mechanisms' they are hypothetical models of success. It would be interesting to infer which of these you believe are occurring in which types of BE in your discussion, but this may be beyond the scope of this review.

Line 63: CRISPR-CRISPR would make the CCas proteins? A mistype of an extra CRISPR I think, please update.

Line 74: I'm torn between knowing that this would be the case (that viruses aid their bacterial hosts in adaptation) because it is completely well documented that they do, and appreciating that this does provide good evidence that this is occurring in BEs. I think you had a unique opportunity here with the data generated to actively understand the extent of uses of ecological success models that phages use, as understanding models such as whether the phage-bacteria pairs are undergoing an arms race dynamic or a fluctuating selection dynamic requires such an in depth look at the defence systems and polymorphisms. Perhaps something to consider for the future to make the most out of this extensive dataset!

The results are well written and describe the data well.

Line 84/85: please include the criteria used for cutoff in VirSorter 2 (which categories were used) and DeepVirFinder (did you use a p-value cutoff?) This is crucial information and is more informative than just stating which tools were used. This is [particularly important when assessing the short contigs from DeepVirFinder, as Kmer-based techniques can pick up many plasmids and other false positives. Although keeping all contigs >1Kb is not beyond the realms of sensible, I think for future reference it can be better to keep mostly longer contigs (>5Kb) and then interrogate the short contigs for viruses that you want to find or expect to find, for example mining using DeepVirFinder trained on an Anelloviridae database if looking for short gastrointestinal associated viruses. I don't think the inclusion of short contigs drastically changes the results here, so no issue. This is particularly true when using bulk metagenome sequence data, as viral enriched samples will have more RNA viruses which will make the size distribution of true positives shift to the left tail slightly.

Line 87: Interesting, for some reason I would expect more viral genomes.

Line 92: perhaps add a small explanation of how you determined host integration for the readers ease.

Line 102: I'm not a statistician, but I would double check that you can use a PERMANOVA here, mainly by checking that you can use the typical dissimilarity measure included in PERMANOVAs for the data included in the analysis, but the stats seem to be pretty conclusive and interpret the results honestly. I think ANOVA has the main issue that assumptions have to be made about the data, make sure that each of your independent variables can have an assumption of no outliers, equal variance, normal distribution and that they are in fact completely independent! I have not seen any writing about attempts to remove or understand covariates from the analysis.

Line 121: again, the order Caudovirales, I'm not sure on the official stance but I believe you will have found tailed bacteriophages, which don't have a single evolutionary origin, therefore probably should not constitute a single order. Be sure to check the ICTV most recent updates. Add a small mention of how you assigned taxonomy here for ease of reading.

Line 126: Fascinating!

Line 130: Terrifying!

Line 183: Very well structured ARG analysis! Many studies do not address different databases.

Line 190: This analysis is robust and very well executed to my understanding, especially due to the difficulty in inferring virus-bacteria links.

Line 277: I think this claim is a bit bold for the evidence provided. You can say that certain viruses encode certain AMGs and link this to hosts and abundance within a host, but I'm not sure you can concretely mention that adaptation of the host was driven by AMGs... Many AMGs may actually just be serving an unknown purpose within the phage lifecycle or as anti-anti-phage producing mechanisms, but we just don't have enough mechanistic information about the genes. I find this limitation is ubiquitous in the literature, and we now have more data about genomes than we do mechanistic understanding of the genes within the data.

The discussion is well written and makes useful and I think honest interpretations of the data.

Line 437: definitely add the parameters used for inclusion/exclusion of viruses from DeepVirFinder and VirSorter 2 here, this information is crucial for reproducibility.

Line 442: protein sharing network (be specific) for VConTACT2

Line 448: Cite MetaWRAP and cite each tool used in MetaWRAP please!

The reference list appears to be complete and consistently formatted.

Figures are of good standard and appropriate for the data.

I thoroughly enjoyed reading this manuscript, and will certainly recommend it to colleagues after publication.

Reviewer #3 (Remarks to the Author):

The paper by Du et al reports the analysis of several hundred viromes from various built environments (BE) including different surfaces from human dwellings as well as human skin. Viromes from BE have not received much attention until now, so the study is timely and potentially important. A number of interesting findings is reported including antimicrobial resistance genes, auxiliary metabolic genes and ACRs in virus genome, distinct biases in host range and more. On the whole the analysis is thorough and done with state of the art methods. That said, there are a number of questions and I think points to clarify and revisit as listed below.

1. It is unclear to me whether the BE microbiomes - and accordingly viromes - represent the microbiota that actually grows in those habitats or mostly contaminations. The authors mention in a few places in the manuscript that these habitats are oligotrophic and generally represent harsh conditions, implying they believe those bacteria grow there. However, this needs to be clarified and discussed because the implications for the interpretation of the results are major.
2. The degree of novelty in the characterized viromes remains rather obscure. Figure 2 is not really helpful. It is not entirely clear how these trees were built (that is, what were the underlying alignments) and what do they include. Only vOTUs identifiable to the species level? I guess so but I am not sure. I think the most sensible approach is to build trees for capsid proteins and large terminase subunits of Caudovirecetes from the analyzed viromes and clearly indicate known and novel clades.
3. The analysis of ACRs is not quite satisfactory. More information should be provided regarding the novel ones - ideally, even including structural modeling using AlphaFold2. Furthermore, I find the tree in Fig. 5c obscure and suspicious. There simply aren't so many homologous ACRs, so I fear the authors built a tree for non-homologous proteins which certainly is not a legitimate practice. If I am mistaken, that could be because the legend is not at all helpful.
4. The authors use outdated virus taxonomy. There are no such taxa now as Caudovirales or Siphoviridae. This should be replaced throughout with the current ICTV taxonomy.
5. The manuscript contains a lot of detail on Caudovirecetes but not much at all on any other viruses. Do those show any distinctive features?

Response to reviewers' comments

We thank the editors and reviewers for their comments on our manuscript. We are extremely grateful for your time and constructive comments, which have helped us significantly improve our manuscript. Please find below our responses to each of the reviewers' comments. The corresponding changes have been shown in blue font, and the relevant line numbers in the clean document have been specified.

REVIEWER COMMENTS

Reviewer #1 (Remarks to the Author):

Du et al. report on a very large dataset of 738 bulk metagenomes from built environments, that they have scrutinized under the angle of their (pro?)phage content. The number of analyses they have performed is quite large as well. However, in terms of precision of the analyses, and of the vocabulary used throughout to describe the results, some difficulties emerge. At this stage, I cannot recommend publication of this manuscript, for the main reasons listed below.

- 1. The authors are probably unfamiliar with phage biology, so the vocabulary is confuse throughout. Some examples: a lysogen designates a bacterium hosting a prophage, and not the prophage itself. Phages “are” not lytic, they make lytic cycles. All phages, whether temperate or virulent, make lytic cycles, otherwise they are defective prophages. A prophage needs not to be integrated in the bacterial genome, it can be maintained as a plasmid. So the absence of an integrase or a junction with bacterial DNA is not sufficient to declare a phage virulent.*

Thank you for your comment. We have revised our vocabulary describing phages and their hosts throughout the manuscript to ensure that the terms used are appropriate. In particular, we have revised the usage of the terms related to lysogenic and lytic cycles. Of all of the prophages identified using the bioinformatics tools PHASTER (Arndt et al., 2016) and VIBRIANT (Kieft et al., 2020), only those that were integrated into a bacterial genome were considered as having high confidence (the prophage region's total score was >90 using PHASTER) and were reported in this study; prophages that were maintained as plasmids were not reported. We have indicated the scope of our prophage analysis in the Methods section of the Supplementary Notes. We understand that the absence of an integrase does not sufficiently indicate that a phage is virulent. In fact, interestingly, we identified one complete circular vOTU with lytic cycles that still harbored an integrase (please see Fig. S10b, vOTU SL336690_c_82_full).

We have made the following changes in the revised manuscript:

[Lines 227–228, 230, 274, 380, and 383] “...viruses with lysogenic cycles...”

[Line 280] “...vOTUs with lysogenic cycles...”

[Line 275] “...viruses that make lytic cycles...”

[Line 281] “...vOTU (SL336690_c_82_full) that makes lytic cycles...”

[Line 778, Figure S7 legend] “...viruses with lysogenic and non-lysogenic cycles...”

[Lines 788–789, Figure S10 legend] “...viral operational taxonomic units (vOTUs) found to make the (a) lysogenic or (b) lytic cycle...”

Supplementary Note [Lines 80–84] “In total, 332 unique proviruses were identified using the two aforementioned tools and CheckV, and only those (127) that were integrated into a bacterial genome had high confidence (the prophage region’s total score was >90 in PHASTER) and were included in the downstream analysis.”

References

Arndt, D. et al. PHASTER: A better, faster version of the PHAST phage search tool. *Nucleic Acids Res.* 44, W16-W21 (2016).

Kieft, K., Zhou, Z. & Anantharaman, K. VIBRANT: automated recovery, annotation and curation of microbial viruses, and evaluation of viral community function from genomic sequences. *Microbiome* 8, 90 (2020).

2. *Linked with this vocabulary problem, there is a deeper problem of understanding what is at stake when a bulk metagenome is analysed for its (pro)phage content. The Supplementary Method section is too superficial to inform on very basic points such as how were the samples lysed? Were spiked phage particles introduced and recovered quantitatively? What is studied really? The prophage content, including all defective prophages that are rampant in bacterial genomes? What is the question addressed finally?*

In line with your suggestions, we have added more details, especially some basic information, regarding the methods in the **Supplementary Note**. Several key points have been summarized below.

- a.) Genomic DNA was extracted from swabs collected from sampled surfaces using the ZymoBIOMICS 96 MagBead DNA kit (Zymo Research, CA, USA), following the manufacturer’s instructions. Cell lysis was performed with a chemical solution and mechanical bead beating according to the kit protocols. Using a customized protocol, genomic DNA was extracted from air sampling filters. In brief, cell lysis was performed using NucliSENS Lysis Buffer (BioMérieux, Marcy-l’Étoile, France) and a multi-enzyme cocktail (MetaPolyzyme, Sigma-Aldrich, MO, USA), followed by mechanical bead beating. Detailed information about the extraction process can be found in the studies by Danko et al. (2021) and Marcus et al. (2021).

Supplementary Note [Lines 40–46] “In brief, genomic DNA was extracted from swabs collected from sampled surfaces using the ZymoBIOMICS 96 MagBead DNA kit (Zymo Research, CA, USA), following the manufacturer’s instructions. The cells were lysed using a chemical solution and mechanical bead beating technology. Genomic DNA was extracted from air sampling filters using a customized protocol. Cell lysis was performed using NucliSENS Lysis Buffer (BioMérieux, Marcy-l’Étoile, France) and a multi-enzyme cocktail (MetaPolyzyme, Sigma-Aldrich, MO, USA), followed by mechanical bead beating.”

- b.) A ZymoBIOMICS synthetic microbial community standard (Zymo Research Corporation, CA, USA), instead of spiked phage particles, was set as the positive control for bulk metagenomic sequencing in the extraction process. The positive control was processed in tandem with the field samples to verify the extraction and

sequencing methodologies. The positive controls yielded the expected sequencing results, indicating the reliability of our bulk metagenomic sequencing methodology.

Supplementary Note [Lines 46–49] “A ZymoBIOMICS synthetic microbial community standard (Zymo Research Corporation, CA, USA) was used as the positive control in the extraction process and sequenced in tandem with the surface and air samples.”

Supplementary Note [Line 55] “The positive controls yielded the expected sequencing results.”

- c.) The main objectives of this study were to analyze the bacteria and viruses present in built environments (BEs) and identify the linkages between viruses and their putative bacterial hosts using culture-independent metagenomic sequencing. To the best of our knowledge, this is the first study to comprehensively characterize viromes and virus–host interactions in BEs. This study also highlights the significance of analyzing bulk metagenomes to study the viromes of BEs, which have important health implications for occupants. We have clarified the main objectives of this study in the Introduction section.

[Lines 72–75] “CRISPR/Cas systems have been reported in surface microbiomes across urban environments worldwide¹⁵; however, the immune mechanisms of infection and the virus–host interactions (e.g., the extent of virus–host links, prevalent viral life cycle, and novelty of Acr proteins) that occur in BEs are poorly understood.”

- d.) As mentioned in our response to Comment #1, only prophages that were integrated into a bacterial genome were included in this study. The identified prophages should include all functional and defective ones because bulk metagenomic sequencing does not allow differentiation between the two. Although advances in genomic sequencing have yielded fundamental insights into the genetic diversity of uncultured viruses and their evolutionary dynamics within hosts in environmental (e.g., terrestrial and aquatic) and host-associated (e.g., plants, animals, and humans) systems (Sommers et al., 2021), bioinformatics approaches suffer from a drawback in that they cannot determine whether the identified prophages are functional or defective. Identifying a functional prophage in a bacterial genome is challenging for the following reasons (Zhao et al., 2010): (i) bacterial genomes may not be fully annotated, and some phage-like open reading frames may therefore be missing; (ii) only a few phage-like genes may be found within a short sequence region; (iii) phage-like genes are spread over a reasonably compact region, but the whole element represents only the remnant of a once-functional prophage that underwent mutational decay and is therefore no longer inducible; and (iv) owing to their enormous diversity, prophages that are yet-undiscovered may be hidden within bacterial genomes that are currently considered fully annotated. We have added this limitation to the Discussion section, describing that defective or functional prophages cannot be differentiated using bulk metagenomic sequencing and stating that future culture-based work will be required.

[Lines 424–427] “Second, the results derived from metagenomic sequencing did not differentiate whether the identified viruses and their associated genes (e.g., ARGs and Acr proteins) were functional or defective. Future experimental investigations will be required to identify the functions of the novel Acr proteins.”

e.) Although functional and defective prophages cannot be differentiated, our results highlight the characteristics of viruses and the virus–host interactions in BEs, which have yet to be reported in the literature. Our results also provide insights for future experimental studies. As mentioned above, we have clarified the research questions in the Introduction section.

[Lines 72–75] “CRISPR/Cas systems have been reported in surface microbiomes across urban environments worldwide¹⁵; however, the immune mechanisms of infection and the virus–host interactions (e.g., the extent of virus–host links, prevalent viral life cycle, and novelty of Acr proteins) that occur in BEs are poorly understood.”

References

Danko, D. et al. A global metagenomic map of urban microbiomes and antimicrobial resistance. *Cell* 184, 3376-3393 (2021).

Leung, M. H. Y., et al. Characterization of the public transit air microbiome and resistome reveals geographical specificity. *Microbiome* 9(1), 1-19 (2021).

Zhao, Y., Wang, K., Ackermann, H.W., Halden, R. U., Jiao, N. & Chen, F. Searching for a “hidden” prophage in a marine bacterium. *Appl. Environ. Microbiol.* 76(2), 589-595 (2010).

Sommers, P., Chatterjee, A., Varsani, A., & Trubl, G. Integrating viral metagenomics into an ecological framework. *Annu. Rev. Virol.* 8(LLNL-JRNL-818035) (2021).

3. *The authors think they found beta-lactamase genes in their prophage genomes. They seem to be aware that antibiotic resistance genes are rare in phage genomes, and that many studies were flawed because of irrelevant BLAST cut-offs (at least they cite papers about it). I could not find any indication of the cut-off they used, and the arguments in favour of bona fide antibiotic resistance genes, present on functional prophages (rather than defective ones).*

We used two bioinformatics tools (i.e., Resistance Gene Identifier and NCBI AMRFinder tool) and two reference databases to identify the presence of antibiotic resistance genes (ARGs) in phages. We have indicated the cut-off and thresholds used in the Methods section of the Supplementary Notes. In particular, the ARGs in the viral genomes were annotated using (1) the Resistance Gene Identifier tool (v.5.2.0) with the option “–low_quality,” which applied the best identity of >60% to the reference sequences in the CARD database (v.3.0.9), and (2) the NCBI AMRFinderPlus tool (v.3.8.4), with default options of 60% coverage and 80% identity, to the reference sequences in the Resfams database. The search was performed using the hmmsearch utility tool in HMMER (v.3.1b2), with an E-value $\leq 1e^{-5}$ and a gathering threshold score ≥ 40 . The Resfams annotation with the best score was adopted when an ARG received different annotations from the databases.

Supplementary Note [Lines 174–181] “The ARGs in the viral genomes were annotated using the Resistance Gene Identifier tool (v.5.2.0)²⁷ with the option “–low_quality,” which applied the best identity of > 60% to the reference sequences in the CARD database (v.3.0.9),²⁸ and the NCBI AMRFinderPlus (v.3.8.4) tool,²⁹ with the default options of 60% coverage and 80% identity, to the reference sequences in the Resfams database³⁰. The search was performed using the hmmsearch utility in the HMMER tool (v.3.1b2), with an E-value $\leq 1e^{-5}$ and a gathering threshold score ≥ 40 . The Resfams annotation with the best score was adopted when an ARG received different annotations from the databases.”

Functional and defective prophages cannot be differentiated using bulk metagenomics sequencing and bioinformatics methods. However, according to published reports, a large fraction of the prophages identified by genomic analysis appear to be defective and inactive (Casjens, et al., 2003 & Canchaya, et al., 2003). Defective prophages may lack the ability to lyse their hosts; however, certain functional genes in these defective prophages could be beneficial to the evolution and ecological fitness of their hosts. We have emphasized in the manuscript that caution should be exercised when understanding ARGs because they are generally rare in prophages and may not be functional. We have also indicated that future studies will need to further investigate whether functional prophages carry antibiotic resistance genes.

[Lines 394–396] “Although phage-mediated ARG transduction is rare and the ARGs may not be functional^{37,68}, antibiotic resistance conferred by functional beta-lactamase-encoding genes has been identified in freshwater viromes and validated experimentally⁶⁹.”

[Lines 402–404] “Thus, the role played by viruses in the development of antibiotic resistance in bacteria, especially those present on frequently touched surfaces in BEs, is crucial and warrants further investigation.”

References

Casjens, S. Prophages and bacterial genomics: what have we learned so far? *Mol. Microbiol.* 49, 277-300 (2003).

Canchaya, C., Proux, C., Fournous, G., Bruttin, A. & Brüssow, H. Prophage genomics. *Microbiol. Mol. Biol. R.* 67(2), 238-276 (2003).

4. *In general, there are too many studies, and not sufficiently well described. I would rather choose some interesting aspects, and describe them to the exact details of how the analysis was done, so that someone else can repeat them. The Acr detection is one of the original aspects of this work.*

Because this is the first comprehensive study to report viromes in BEs, we presented the diversity, taxonomy, and function of viruses in the first two sections of the manuscript to provide the readers an effective overview. In the subsequent sections, we have emphasized the interactions between phages and their bacterial hosts in BEs and how these interactions could benefit the evolution and ecological fitness of the hosts; these are the most significant findings of this study. As indicated by you, the detection of Acr proteins is another key significant finding of this study. We have revised the Introduction section to highlight the key research questions that we have addressed in this manuscript. We have also amended the Results section related to Acr proteins to highlight the novel ones identified.

To ensure that the other researchers can reproduce our results, we have made all of the codes used in the data analysis publicly available on GitHub (https://github.com/dscdorothy/HK_BE_viromic).

[Lines 72–75] “CRISPR/Cas systems have been reported in surface microbiomes across urban environments worldwide¹⁵; however, the immune mechanisms of infection and the virus–host interactions (e.g., the extent of virus–host links, prevalent viral life cycle, and novelty of Acr proteins) that occur in BEs are poorly understood.”

[Lines 263–264] “Nine of the 162 predicted Acr proteins were considered novel, i.e., they did not match a known reference with a BLAST-based hit E-value < 0.001 (Table S10).”

[Lines 266–270] “Computational modeling of all the predicted Acr proteins revealed diverse structures (48 were considered as having high confidence [pI_{DDT} > 80]) (Fig. S9). Comparison of the high-confidence structures of the four novel predicted Acr proteins with their closest references revealed differences, which may be responsible for the variations in their functions (Fig. 5d).”

[Lines 479–480] “The high-confidence structures of the predicted Acr proteins are provided at https://github.com/dscdorothy/HK_BE_viroomic.”

[Lines 484–485] “The supporting code is provided at https://github.com/dscdorothy/HK_BE_viroomic and <https://doi.org/10.1038/s41564-021-00928-6>.”

Reviewer #2 (Remarks to the Author):

Shicong Du and co-authors have written a very compelling and well interpreted manuscript that appears to be statistically sound and is of very good quality in general. The study provides excellent data further developing our understanding of the diversity of viruses in built environments, sampling from diverse sources, and the viromics analysis uses mostly up to date techniques. This being said, there is very little in the way of discussion on the various biases that can occur at each stage in the viromics process, and this type of information and self reflection is fundamental to effective interpretation of metagenomic virus data. The findings are indeed of significant interest to the viromics community, particularly the discovery of AMR genes in different host-specific built environments, and the CRISPR/anti-CRISPR led virus-host link analysis is particularly striking for reasons such as novelty and ingenuity.

Thank you for your comment. We acknowledge that potential biases could have occurred in our virome analysis workflow. We have discussed the potential biases that can occur at each stage of the virome analysis process. Specifically, we have highlighted that the diversity of viruses and their hosts could be underestimated owing to the low biomass in the BE samples and the potential biases associated with the DNA extraction and purification methods. Furthermore, the lack of appropriate references in the bioinformatics tool databases could lead to poorly characterized sequences (Callanan et al., 2021).

[Lines 419–424] “First, potential biases could occur in our virome analysis workflow. Owing to the low biomass in the BE samples and the potential biases associated with the DNA extraction and purification methods, the diversity of viruses and their hosts could be underestimated. Furthermore, the lack of appropriate references in the bioinformatics tool databases could lead to poorly characterized sequences⁷².”

References

Callanan, J., Stockdale, S. R., Shkoporov, A., Draper, L. A., Ross, R. P., & Hill, C. Biases in viral metagenomics-based detection, cataloguing and quantification of bacteriophage genomes in human faeces, a review. *Microorganisms* 9(3), 524 (2021).

I have the following comments on the manuscript:

Line 27: I think ‘Caudovirales’ can be a bit triggering for any viral taxonomists, as tailed bacteriophages appear to have more than one origin, I suggest sticking to ‘tailed bacteriophages’ if this is specifically what you are referring to, which I believe it is.

We acknowledge that “Caudovirales” is an outdated virus taxonomy according to the most updated version of the database published by the International Committee on Taxonomy of Viruses (ICTV) (<https://ictv.global/taxonomy>). Accordingly, we have re-analyzed the taxonomy of all viruses using the latest version of IMG/VR (v.4; released 2022-09-20) (Camargo et al., 2023) and revised the taxa names. All of the corresponding figures (Figs. 1d and S3) and tables (Tables S4 and S10) have been updated with the correct taxonomy. We have also referred to “Caudovirales” as “tailed bacteriophages” where appropriate.

[Lines 120–132] “Most of the vOTUs (92.4%) could not be taxonomically classified into a known viral genus or family, similar to the reported rate of novelty for the viromes collected from other ecosystems^{2,30}, and could only be resolved as unclassified members of the class Caudoviricetes (Fig. 1c). Among the annotated vOTUs, 1.7%, 1.7%, 1.3%, and 0.6% belonged to the dsDNA viral genus

Pahexavirus, ssRNA-RT viral family Metaviridae, ssDNA viral family Genomoviridae, and ssDNA viral family Inoviridae, respectively (Fig. S3). Specifically, the members of the family Autographiviridae, an extensively studied family of virulent phages³¹, were enriched and dominant in subway air (Fig. S3, Table S4); in contrast, the members of the genus *Pahexavirus* were abundant on doorknobs and skin surfaces (Fig. S3, Table S4), which is not surprising because these have been shown to infect skin bacteria (e.g., *Propionibacterium*³²). Furthermore, the members of the order Orthopolintovirales, an emerging group of viruses known as virophages³³, were also enriched on human skin-associated surfaces (Fig. S3).”

[Lines 138–139] “Caudoviricetes, a class of viruses with a helical tail and icosahedral capsid (tailed bacteriophages), is prevalent in diverse ecosystems^{2,30}.”

[Lines 142–145] “The tree showed that the Caudoviricetes vOTUs that were widely distributed in different BE habitats but could not be classified through the IMG/VR database belonged to the genus *Bendigovirus*, whereas a distinct clade comprising one unknown vOTU belonged to the genus *Oshimavirus* (Fig. 2a).”

Figure 1d, Figure S3, Table S4, and Table S10 have been revised with the updated taxonomy.

Reference

Camargo, A. P., et al. IMG/VR v4: an expanded database of uncultivated virus genomes within a framework of extensive functional, taxonomic, and ecological metadata. *Nucleic Acids Res.* 51, D733-D743 (2023).

The introduction focusses a lot on virus-host ecology, and although this makes a fair share of the results, it does not reflect the literature completely fairly. By this, I mean that adding a little more on comparisons between the virome of very different environments and the interpretation (that they are seemingly individualised, and any other interesting points found by the wealth of viromics studies) may improve the introduction to the topic. For a communication, including more on viromics studies and their interpretations of the global virome and it's role may be of more interest and use than explaining the lifestyle of phages, for example. I appreciate that phage lifestyle is also integral to the introduction of phages as ubiquitous entites that are also prevelant in BEs, I would suggest trying to include both what you have written and suggested extras.

We have modified the Introduction section by including additional information about viruses in BEs to better orient the readers before transitioning to the virus–host ecology. We have also briefly described the viromes in other environmental systems in the Introduction section.

[Lines 40–45] “The total concentration of the viruses in BEs is estimated to be approximately 10⁵ particles/cubic meter⁸. Although the environmental conditions of most BEs are oligotrophic and considered poorly suited for microbial life⁹, a conspicuous diversity of viruses, including epidemic-associated viruses (e.g., SARS-CoV-2¹⁰ and yellow fever virus¹¹), have been found in microbial communities in air and on surfaces in BEs.”

[Lines 45–47] “A few studies on viromes in public buildings (e.g., daycare centers and restrooms) have mainly focused on a small spatial scale and limited sample types and have not investigated the bacterial hosts of the viruses¹²⁻¹⁴.”

[Lines 57–60] “In most marine and soil environments, phages are often highly diverse and abundant, thereby routinely infecting a significant fraction of their microbial hosts, which, together with the

expression of virus-encoded auxiliary metabolic genes (AMGs) in host genomes, plays a key role in global nutrient cycling^{4,20,21}.”

(General comment: please change all ‘Bes’ to BE to make the acronym consistent throughout)

We have consistently revised to BEs throughout the manuscript.

Line 60: please see up to date inference of the different models of phage success (<https://www.sciencedirect.com/science/article/pii/S1369527422001138><[https://urldefense.c om/v3/ https://www.sciencedirect.com/science/article/pii/S1369527422001138 ;!!KjDnqvt InNPT!mlJzSSWnQal8os1gYT8a9FUmmhHWta87nt0iu8Z3gdWA68PYllawmtJwRINi5-9PzlyaKeNqzd3EgXHy6vtQPRFEEM9VCIir9w\\$>](https://urldefense.c om/v3/ https://www.sciencedirect.com/science/article/pii/S1369527422001138 ;!!KjDnqvt InNPT!mlJzSSWnQal8os1gYT8a9FUmmhHWta87nt0iu8Z3gdWA68PYllawmtJwRINi5-9PzlyaKeNqzd3EgXHy6vtQPRFEEM9VCIir9w$>)). For example, if one understands these hypotheses then one appreciates that the ‘Royal family’ model is actually just different kill-the-winner models, and in fact one of these models in the Royal Family can be explained by an arms race dynamic or fluctuating selection dynamic. Also, I would not refer to these as ‘mechanisms’ they are hypothetical models of success. It would be interesting to infer which of these you believe are occurring in which types of BE in your discussion, but this may be beyond the scope of this review.

As indicated, we have updated the references for the different ecological models of phage success. We have also replaced the term “mechanisms” with “ecological models.” Moreover, in the Discussion section, we have mentioned that the ecological model viruses use for establishing themselves in the BEs should be investigated in the future.

[Lines 60–63] “From an ecological perspective, phages in a microbial community can mediate the competition among bacterial species by establishing lytic infection through several well-established ecological models, including the “kill-the-winner” and “piggyback-the-winner” models²².”

[Lines 388–390] “However, future studies are required to determine the ecological model viruses use for establishing themselves in the BEs.”

Line 63: CRISPR-CRISPR would make the CCas proteins? A mistype of an extra CRISPR I think, please update.

We have revised the term to CRISPR/CRISPR-associated (Cas) proteins throughout the manuscript.

Line 74: I’m torn between knowing that this would be the case (that viruses aid their bacterial hosts in adaptation) because it is completely well documented that they do, and appreciating that this does provide good evidence that this is occurring in BEs. I think you had a unique opportunity here with the data generated to actively understand the extent of uses of ecological success models that phages use, as understanding models such as whether the phage-bacteria pairs are undergoing an arms race dynamic or a fluctuating selection dynamic requires such an in depth look at the defence systems and polymorphisms. Perhaps something to consider for the future to make the most out of this extensive dataset!

Analyzing the extent of the use of ecological success models used by phages is beyond the scope of this study. As suggested, in the Discussion section, we have mentioned that this topic should be investigated in the future.

[Lines 388–390] “However, future studies are required to determine the ecological model viruses use for establishing themselves in the BEs”

The results are well written and describe the data well.

Line 84/85: please include the criteria used for cutoff in VirSorter 2 (which categories were used) and DeepVirFinder (did you use a p-value cutoff?) This is crucial information and is more informative than just stating which tools were used. This is [particularly important when assessing the short contigs from DeepVirFinder, as Kmer-based techniques can pick up many plasmids and other false positives. Although keeping all contigs >1Kb is not beyond the realms of sensible, I think for future reference it can be better to keep mostly longer contigs (>5Kb) and then interrogate the short contigs for viruses that you want to find or expect to find, for example mining using DeepVirFinder trained on an Anelloviridae database if looking for short gastrointestinal associated viruses. I don't think the inclusion of short contigs drastically changes the results here, so no issue. This is particularly true when using bulk metagenome sequence data, as viral enriched samples will have more RNA viruses which will make the size distribution of true positives shift to the left tail slightly.

We have mentioned the cutoff criteria set for the VirSorter 2 (all categories were used) and DeepVirFinder (p -value < 0.05) tools in the main Methods section and the Methods section in the Supplementary Notes.

[Lines 454–456] “Viral genomes were generated using two tools [i.e., VirSorter2²⁷ (with all categories) and DeepVirFinder²⁸ (v.1.0, with a p -value < 0.05)], and the genome quality was assessed using CheckV (v.0.8.1; database v.1.0)⁷⁶ (Table S2).”

Supplementary Note [Lines 66–68] “The average viral recall of the genomic fragments of different lengths was calculated using VirSorter2⁶ (all categories) with the default cutoff and DeepVirFinder (v.1.0)⁷ with different cut-offs (0.5, 0.7, 0.8, and 0.9) (Fig. S1b).”

Supplementary Note [Lines 69–72] “Based on the mock dataset results, VirSorter2 with the default cutoff (all categories) and DeepVirFinder with a predicated score ≥ 0.5 and a p -value < 0.05, which resulted in the highest viral recall for each fragment length, were used in tandem to process the 4.5 million contigs assembled from 738 metagenome samples (Fig. S1b).”

Line 87: Interesting, for some reason I would expect more viral genomes.

Thank you for the appreciation. We speculate that the environmental conditions in BEs may not favor a large number of viruses and their hosts.

Line 92: perhaps add a small explanation of how you determined host integration for the readers ease.

We have described that host integration was determined based on the provirus integration sites (i.e., host region was predicted on both ends of the viral genome) in detail in the Methods section of the Supplementary Notes (Lines 75–81) as well as in the main text.

[Lines 94–97] “Despite analyzing the bulk metagenomes, only 28% of the viral genomes showed evidence of host integration based on an assessment of the provirus integration sites (i.e., the host region was predicted on both ends of viral genomes) (Fig. 1c).”

Line 102: I'm not a statistician, but I would double check that you can use a PERMANOVA here, mainly by checking that you can use the typical dissimilarity measure included in PERMANOVAs for the data included in the analysis, but the stats seem to be pretty conclusive and interpret the results honestly. I think ANOVA has the main issue that assumptions have to be made about the data, make sure that each of your independent variables can have an assumption of no outliers, equal variance, normal distribution and that they are in fact completely independent! I have not seen any writing about attempts to remove or understand covariates from the analysis.

PERMANOVA is commonly used for dissimilarity analysis to determine the type of datasets used in a study. Other studies (Galand et al., 2018; Emerson et al., 2018; de Jonge et al., 2022) analyzing datasets of a similar nature as ours have also used a similar statistical test.

References

Galand, P.E., Pereira, O., Hochart, C. et al. A strong link between marine microbial community composition and function challenges the idea of functional redundancy. *ISME J.* 12, 2470-2478 (2018).

Emerson, J.B., Roux, S., Brum, J.R. et al. Host-linked soil viral ecology along a permafrost thaw gradient. *Nat. Microbiol.* 3, 870-880 (2018).

de Jonge, P.A., Wortelboer, K., Scheithauer, T.P.M. et al. Gut virome profiling identifies a widespread bacteriophage family associated with metabolic syndrome. *Nat. Commun.* 13, 3594 (2022).

Line 121: again, the order Caudovirales, I'm not sure on the official stance but I believe you will have found tailed bacteriophages, which don't have a single evolutionary origin, therefore probably should not constitute a single order. Be sure to check the ICTV most recent updates. Add a small mention of how you assigned taxonomy here for ease of reading.

As indicated above, we have updated the taxonomy throughout the manuscript and have described how the taxonomy was assigned in the Methods section of the Supplementary Notes (Lines 93–104).

Line 126: Fascinating!

Thank you for the appreciation.

Line 130: Terrifying!

Thank you for your comment.

Line 183: Very well structured ARG analysis! Many studies do not address different databases.

We are glad to know that you appreciate our effort.

Line 190: This analysis is robust and very well executed to my understanding, especially due to the difficulty in inferring virus-bacteria links.

Thank you for the appreciation.

Line 277: I think this claim is a bit bold for the evidence provided. You can say that certain viruses encode certain AMGs and link this to hosts and abundance within a host, but I'm not sure you can concretely mention that adaptation of the host was driven by AMGs... Many AMGs may actually just be serving an unknown purpose within the phage lifecycle or as anti-anti-phage producing mechanisms, but we just don't have enough mechanistic information about the genes. I find this limitation is ubiquitous in the literature, and we now have more data about genomes than we do mechanistic understanding of the genes within the data.

We have revised the subsection's title to "The AMGs of viruses are linked to specific hosts" to avoid over-interpretation of the implications of our findings.

[Line 285] "The AMGs of viruses are linked to specific hosts"

The discussion is well written and makes useful and I think honest interpretations of the data.

Thank you for the appreciation.

Line 437: definitely add the parameters used for inclusion/exclusion of viruses from DeepVirFinder and VirSorter 2 here, this information is crucial for reproducibility.

We have added the inclusion/exclusion criteria for viruses from the DeepVirFinder and VirSorter 2 in the main text and in the Methods section of the Supplementary Notes.

[Lines 454–456] "Viral genomes were generated using two tools [i.e., VirSorter2²⁷ (with all categories) and DeepVirFinder²⁸ (v.1.0, with a p -value < 0.05)], and the genome quality was assessed using CheckV (v.0.8.1; database v.1.0)⁷⁶ (Table S2)."

Supplementary Note [Lines 66–68] "The average viral recall of the genomic fragments of different lengths was calculated using VirSorter2⁶ (all categories) with the default cutoff and DeepVirFinder (v.1.0)⁷ with different cut-offs (0.5, 0.7, 0.8, and 0.9) (Fig. S1b)."

Supplementary Note [Lines 69–72] "Based on the mock dataset results, VirSorter2 with the default cutoff (all categories) and DeepVirFinder with a predicated score of ≥ 0.5 and a p -value of <0.05, which resulted in the highest viral recall for each fragment length, were used in tandem to process the 4.5 million contigs assembled from 738 metagenome samples (Fig. S1b)."

Line 442: protein sharing network (be specific) for VConTACT2

The results derived from vConTACT2 have been deleted because VConTACT2 used an outdated taxonomy in its database. We have removed the corresponding figure (Figure S3 previously) and have adjusted the figure numbering accordingly.

Supplementary Note [Line 96] We have deleted the sentence "vConTACT2 (v.0.9.22) was used to automate the network-based classification of prokaryotic viruses based on shared protein-coding gene clusters between viral genomes¹⁴."

Supplementary Note [Line 98] We have deleted the sentences "...grouped into clusters via all-to-all BLASTp with the RefSeq207 viral genomes as references¹⁴. The degree of similarity between vOTUs was calculated based on the number of shared clusters, and then pairs of closely related vOTUs with a similarity score of ≥ 1 were grouped into the same viral cluster. The gene-sharing networks were

visualized using Cytoscape (v.3.9.0)¹⁶. For vOTUs that could not be clustered with any reference genome,..."

[Line 460] We have deleted the sentence "...network classification using vConTACT2 (v.0.9.22)³⁰ and..."

Line 448: Cite MetaWRAP and cite each tool used in MetaWRAP please!

We have added the citations for both MetaWRAP and each tool within it.

Supplementary Note [Lines 209–211] "The resulting MAGs were further refined using the "bin_refinement" function of MetaWRAP³⁶ and dereplicated using the "dRep dereplicate" function of dRep (v.3.2.2)³⁷."

The reference list appears to be complete and consistently formatted.

Thank you for the appreciation.

Figures are of good standard and appropriate for the data.

Thank you for the appreciation.

I thoroughly enjoyed reading this manuscript, and will certainly recommend it to colleagues after publication.

Thank you! We are glad to learn that you enjoyed reading our manuscript.

Reviewer #3 (Remarks to the Author):

The paper by Du et al reports the analysis of several hundred viromes from various built environments (BE) including different surfaces from human dwellings as well as human skin. Viromes from BE have not received much attention until now, so the study is timely and potentially important. A number of interesting findings is reported including antimicrobial resistance genes, auxiliary metabolic genes and ACRs in virus genome, distinct biases in host range and more. On the whole the analysis is thorough and done with state of the art methods. That said, there are a number of questions and I think points to clarify and revisit as listed below.

- 1. It is unclear to me whether the BE microbiomes - and accordingly viromes - represent the microbiota that actually grows in those habitats or mostly contaminations. The authors mention in a few places in the manuscript that these habitats are oligotrophic and generally represent harsh conditions, implying they believe those bacteria grow there. However, this needs to be clarified and discussed because the implications for the interpretation of the results are major.*

Previous culture-based studies have reported that some of the bacteria and fungi in BEs are active (Angenent et al., 2005; Adams et al., 2017). However, the culture-independent metagenomic sequencing approach used in this study does not allow one to determine whether the bacteria are active and whether the viruses are functional or defective. Nonetheless, the results of this study provide novel insights for future experimental studies. We have added a limitation of this study to the Discussion section: the metabolic status of bacteria and viruses cannot be determined using bulk metagenomic sequencing, and future culture-based studies are thus required. We have also indicated in the Discussion section that the ecological model viruses use for establishing themselves in oligotrophic BEs should be investigated in the future.

[Lines 424–427] “Second, the results derived from metagenomic sequencing did not differentiate whether the identified viruses and their associated genes (e.g., ARGs and Acr proteins) were functional or defective. Future experimental investigations will be required to identify the functions of the novel Acr proteins.”

[Lines 388–390] “However, future studies are required to determine the ecological model viruses use for establishing themselves in the BEs.”

References

Angenent, L.T., Kelley, S.T., St Amand, A., Pace, N.R., Hernandez, M.T. Molecular identification of potential pathogens in water and air of a hospital therapy pool. *Proc. Natl. Acad. Sci. U. S. A.* 102, 4860-4865 (2005).

Adams, R. I. et al. Microbes and associated soluble and volatile chemicals on periodically wet household surfaces. *Microbiome* 5, 128 (2017).

- 2. The degree of novelty in the characterized viromes remains rather obscure. Figure 2 is not really helpful. It is not entirely clear how these trees were built (that is, what were the underlying alignments) and what do they include. Only vOTUs identifiable to the species level? I guess so but I am not sure. I think the most sensible approach is to build trees for capsid proteins and large terminase subunits of Caudovirecetes from the analyzed viromes and clearly indicate known and novel clades.*

The trees shown in Figure 2 were constructed as described previously (Low et al., 2019). The relevant details are provided in the Methods section of the Supplementary Notes (Lines 146–161). We have also briefly described the method in the main text (Lines 138–141). To construct the trees, the 77 Caudoviricetes markers, which include capsid proteins and large terminase subunits (please see the list below in Table 1; the capsid proteins and large terminase subunits are shown in bold), were searched against the protein-coding gene sequences of the vOTUs using a profile hidden Markov model (HMM). The dataset from this study contained 444 vOTUs with lengths ≥ 5 kb and 37,775 protein-coding gene sequences, whereas the three public datasets contained 2,389 vOTUs with 168,256 viral protein-coding gene sequences with lengths ≥ 5 kb. The top HMM hits were individually aligned to the profile HMMs of the 77 markers using the “hmmsearch” function, which retained 40 and 58 markers from this study and the three public datasets, respectively. All of the vOTUs that had a valid hit against the Caudoviricetes markers are shown in Figure 2.

We have highlighted the novel clades in Figure 2 using different colors and clarified the relevant figure legend to make it more informative.

[Lines 139–142] “To investigate whether the members of Caudoviricetes that were present across the BE habitats possess a common evolutionary origin, a phylogenetic tree containing 87 species-level vOTUs was constructed using 77 reference protein-coding marker genes (Fig. 2a).”

[Lines 693–697, Figure 2 legend] “Phylogenomics of the viral operational taxonomic units (vOTUs) from the class Caudoviricetes in the built environments (BEs). (a) A phylogenetic tree of 87 species-level vOTUs derived from BEs. The branches of the two clusters of *Oshimavirus* and *Bendigovirus* are shown in light green and orange, respectively. The gray and white circles denote the novel Caudoviricetes viruses. (b) A phylogenetic tree of 599 species-level vOTUs derived from BEs and other datasets (SMGC, GOV, and MetaSUB). Branches of the novel Caudoviricetes vOTUs are shown in red. Lineages with branch lengths < 0.5 kb were collapsed into a clade and are shown in white on the outer ring.”

Table 1. The 77 Caudoviricetes markers used for constructing the phylogenetic trees shown in Figure 2.

VOGid	Description	VOGid	Description
VOG00022	Integrase	VOG00478	Protein D14
VOG00024	DNA polymerase	VOG00524	SPBc2 prophage-derived uncharacterized protein YonJ
VOG00052	Head completion protein gp16	VOG00574	Gene 5 protein
VOG00053	DNA ligase	VOG00667	Baseplate wedge protein gp6
VOG00057	DNA primase	VOG00671	Terminase, large subunit gp19
VOG00061	Uncharacterized protein near the lysin gene (fragment)	VOG00687	Probable portal protein
VOG00080	Ribonucleoside-diphosphate reductase subunit alpha	VOG00688	Head completion protein Gp50
VOG00091	Deoxycytidylate deaminase	VOG00799	Portal protein gp20
VOG00094	Gene 69 protein	VOG00813	Minor tail protein Gp28

VOG00100	Ribonucleoside-diphosphate reductase large subunit	VOG00885	Neck protein gp13
VOG00108	ATP-dependent helicase 41	VOG00917	Hypothetical protein
VOG00110	Baseplate protein J	VOG00977	Uncharacterized 17.5-kDa protein in tk-vs intergenic region
VOG00122	Gene 22 protein	VOG01018	Minor tail protein Gp27
VOG00123	Exodeoxyribonuclease	VOG01027	ATP-dependent DNA helicase uvsW
VOG00143	Baseplate wedge protein gp25	VOG01065	Peptidoglycan hydrolase gp16
VOG00148	Hypothetical protein	VOG01068	Endolysin B
VOG00154	Gene 19 protein	VOG01103	Probable major capsid protein gp17
VOG00180	Major tail protein Gp23	VOG01124	Recombination protein uvsY
VOG00206	DNA-directed DNA polymerase	VOG01126	Spanin, inner membrane subunit
VOG00239	Sliding-clamp-loader gp44 subunit	VOG01137	Deoxynucleotide monophosphate kinase
VOG00246	Putative protein p41	VOG01286	Gene 54 protein
VOG00254	Ribonucleoside-diphosphate reductase subunit beta	VOG01550	Gene 15 protein
VOG00269	Minor capsid protein 10B	VOG01564	T7 RNA polymerase
VOG00282	Minor head protein GP7	VOG01638	Tail tubular protein gp11
VOG00318	Gene 31 protein	VOG01666	Recombination and repair protein
VOG00349	Antirepressor protein ant	VOG01872	Probable capsid assembly scaffolding protein
VOG00363	Tail sheath protein	VOG02305	Gene 32 protein
VOG00402	Exonuclease subunit 2	VOG02317	Gene 30 protein
VOG00413	Internal virion protein gp14	VOG02621	Single-stranded DNA-binding protein
VOG00439	Baseplate central spike complex protein gp27	VOG03123	Single-stranded DNA-binding protein
VOG00450	Prohead core protein protease	VOG03134	Portal protein
VOG00462	Portal protein	VOG03197	Ribonuclease H
VOG00629	Helix-destabilizing protein	VOG03229	PhoH-like protein
VOG03385	Tail completion protein gp15	VOG03444	Prohead protease
VOG03743	Probable flavin-dependent thymidylate synthase	VOG09266	Endonuclease I
VOG12699	Terminase, large subunit	VOG12984	RNA polymerase sigma factor
VOG17540	Minor tail protein	VOG18218	Gene 31 protein
VOG18677	Capsid assembly scaffolding protein	VOG20138	Tail tubular protein gp12
VOG21972	Gene 58 protein		

Reference

Low, S. J., Džunková, M., Chaumeil, P. A., Parks, D. H. & Hugenholtz, P. Evaluation of a concatenated protein phylogeny for classification of tailed double-stranded DNA viruses belonging to the order Caudovirales. *Nat. Microbiol.* 4, 1306-1315 (2019).

3. *The analysis of ACRs is not quite satisfactory. More information should be provided regarding the novel ones - ideally, even including structural modeling using AlphaFold2. Furthermore, I find the tree in Fig. 5c obscure and suspicious. There simply aren't so many homologous ACRs, so I fear the authors built a tree for non-homologous proteins which certainly is not a legitimate practice. If I am mistaken, that could be because the legend is not at all helpful.*

Details regarding the construction of the Acr trees are provided in the Methods section (Lines 463–465). The Acr tree shown in Figure 5c includes 339 known Acr proteins obtained from published studies (compiled using PaCRISPR) as references and 162 predicted Acr proteins that were identified in this study. All of the Acr proteins identified in this study were dereplicated, and only the non-redundant ones were retained for downstream analyses. We have indicated that nine predicted Acr proteins were novel (did not match a known reference with a BLAST-based hit E-value < 0.001); these proteins are shown in red in the tree presented in Figure 5c. The distribution of the predicted Acr proteins in the different viruses is shown in Table S10. We have also modeled the structures of all of the predicted Acr proteins using AlphaFold2. All of the modeled structures of the predicted Acr proteins with a high confidence score (pLDDT > 80) are shown in Figure S9. A comparison of the high-confidence structures of the novel predicted Acr proteins against their closest reference is shown in Figure 5d.

We have accordingly revised the figure legend of Figure 5c to make it more informative.

[Lines 263–273] “Nine of the 162 predicted Acr proteins were considered novel, i.e., they did not match a known reference with a BLAST-based hit E-value < 0.001 (Table S10). A phylogenetic tree was constructed to further determine the uniqueness of all of the predicted Acr proteins, which were found to be broadly distributed in different sub-types (Fig. 5c). Computational modeling of all of the predicted Acr proteins revealed diverse structures (48 were considered as having high confidence [pLDDT > 80]) (Fig. S9). Comparison of the high-confidence structures of the four novel predicted Acr proteins with their closest references revealed differences, which may be responsible for the variations in their functions (Fig. 5d). Several predicted Acr proteins were located in complete circular vOTUs (with lysogenic and lytic cycles) of unknown families (Fig. S10a-b), suggesting that these proteins play roles in the evolution of poorly characterized Caudoviricetes viruses.”

[Lines 464–467] A phylogenetic tree of Acr proteins was constructed using reference sequences from the curated PaCRISPR database⁴⁸ as well as candidate sequences from this study, as described previously⁷⁹. The structures of the Acr proteins were predicted using the AlphaFold2 tool with the default settings^{81,82}.

[Lines 479–480] “The high-confidence structures of the predicted Acr proteins are provided at https://github.com/dscdorothy/HK_BE_viroomic.”

[Lines 726–732, Figure 5 legend] “(c) A phylogenetic tree of the 162 Acr proteins identified in this study and the 339 Acr proteins obtained from the curated PaCRISPR database⁴⁸. The branches of the nine novel predicted Acr proteins are shown in red. The red stars on the outer ring denote the predicted Acr proteins with high-confidence structures (pLDDT > 80) based on the AlphaFold2

tool. (d) Comparison of the high-confidence structures (pI-DDT > 80) of the four novel predicted Acr proteins with their closest reference.”

4. *The authors use outdated virus taxonomy. There are no such taxa now as Caudovirales or Siphoviridae. This should be replaced throughout with the current ICTV taxonomy.*

We acknowledge that “Caudovirales” is an outdated virus taxonomy according to the most updated version of the database published by the International Committee on Taxonomy of Viruses (ICTV) (<https://ictv.global/taxonomy>). Accordingly, we have re-analyzed the taxonomy of all viruses using the latest version of IMG/VR (v.4; released 2022-09-20) (Camargo et al., 2023) and revised the taxa names. All of the corresponding figures (Figs. 1d and S3) and tables (Tables S4 and S10) have been updated with the correct taxonomy. We have also referred to “Caudovirales” as “tailed bacteriophages” where appropriate.

[Lines 120–132] “Most of the vOTUs (92.4%) could not be taxonomically classified into a known viral genus or family, similar to the reported rate of novelty for the viromes collected from other ecosystems^{2,30}, and could only be resolved as unclassified members of the class Caudoviricetes (Fig. 1c). Among the annotated vOTUs, 1.7%, 1.7%, 1.3%, and 0.6% belonged to the dsDNA viral genus *Pahexavirus*, ssRNA-RT viral family Metaviridae, ssDNA viral family Genomoviridae, and ssDNA viral family Inoviridae, respectively (Fig. S3). Specifically, the members of the family Autographiviridae, an extensively studied family of virulent phages³¹, were enriched and dominant in subway air (Fig. S3, Table S4); in contrast, the members of the genus *Pahexavirus* were abundant on doorknobs and skin surfaces (Fig. S3, Table S4), which is not surprising because these have been shown to infect skin bacteria (e.g., *Propionibacterium*³²). Furthermore, the members of the order Orthopolintovirales, an emerging group of viruses known as virophages³³, were also enriched on human skin-associated surfaces (Fig. S3).”

[Lines 138–139] “Caudoviricetes, a class of viruses with a helical tail and icosahedral capsid (tailed bacteriophages), is prevalent in diverse ecosystems^{2,30}.”

[Lines 142–145] “The tree showed that the Caudoviricetes vOTUs that were widely distributed in different BE habitats but could not be classified through the IMG/VR database belonged to the genus *Bendigovirus*, whereas a distinct clade comprising one unknown vOTU belonged to the genus *Oshimavirus* (Fig. 2a).”

Figure 1d, Figure S3, Table S4, and Table S10 have been revised with the updated taxonomy.

Reference

Camargo, A. P., et al. IMG/VR v4: an expanded database of uncultivated virus genomes within a framework of extensive functional, taxonomic, and ecological metadata. *Nucleic Acids Res.* 51, D733-D743 (2023).

5. *The manuscript contains a lot of detail on Caudoviricetes but not much at all on any other viruses. Do those show any distinctive features?*

In this study, more than 70% of the identified viruses were members of Caudoviricetes, which is the main reason why we devoted significant attention to this virus. We have also focused on the members of Orthopolintovirales, an emerging group of viruses known as virophages (Duponchel & Fischer, 2019), which are enriched on skin-associated surfaces.

[Lines 130–132] “Furthermore, the members of the order Orthopolintovirales, an emerging group of viruses known as virophages³³, were also enriched on human skin-associated surfaces (Fig. S3).”

Reference

Duponchel, S. & Fischer, M. G. Viva lavidaviruses! Five features of virophages that parasitize giant DNA viruses. *PLoS Pathogens* 15: e1007592 (2019).

Reviewer comments, second round –

Reviewer #1 (Remarks to the Author):

The answers to the various points raised are honest and satisfactory, so the revised manuscript is acceptable for me.

Reviewer #3 (Remarks to the Author):

The authors have made substantial effort to revise the paper and clarify the results. Yet, I still have some remaining questions.

Regarding the phage tree in Fig. 2, the procedure is now explained in detail. However, regrettably, this poses new questions. The authors used 77 phage gene markers, but report no topology compatibility analysis for the individual gene phylogenies. Without such an analysis, it is hard to trust the tree because horizontal gene transfer among phages is extensive, making phage genomes mosaic. It might be better to analyze a smaller number of highly conserved markers, ensuring compatibility.

With regard to the ACR tree in Fig. 5, I still cannot understand what exactly was done to obtain this phylogeny.

All the above apart, Reviewer 1 raises a variety of pertinent questions on phage biology, and I am not sure these are fully addressed in the revision.

Response to reviewers' comments

We thank the reviewers for their comments on our manuscript. We are extremely grateful for your time and constructive comments, which have helped us significantly improve our manuscript. Please find below our responses to each of the reviewers' comments. The corresponding changes have been shown in blue font, and the relevant line numbers in the clean document have been specified.

REVIEWER COMMENTS

Reviewer #1 (Remarks to the Author):

The answers to the various points raised are honest and satisfactory, so the revised manuscript is acceptable for me.

Thank you for the comment. We are glad to know that our revised manuscript is of a satisfactory standard to you.

Reviewer #3 (Remarks to the Author):

The authors have made substantial effort to revise the paper and clarify the results. Yet, I still have some remaining questions.

Regarding the phage tree in Fig. 2, the procedure is now explained in detail. However, regrettably, this poses new questions. The authors used 77 phage gene markers, but report no topology compatibility analysis for the individual gene phylogenies. Without such an analysis, it is hard to trust the tree because horizontal gene transfer among phages is extensive, making phage genomes mosaic. It might be better to analyze a smaller number of highly conserved marker, ensuring compatibility.

Thank you for the comment. We would like to clarify that the 77 Caudoviricetes gene markers that were used as references to construct the phylogenetic trees shown in Fig. 2 were curated by Low et al. (2019). The authors showed that these gene markers were highly consistent with phage lifestyle and evolutionary mode (and to a lesser extent with genome size of viruses). The same set of phylogenetic markers has been used in a seminal work that studied viruses in the human gut microbiome by Nayfach et al. (2021). Our tree construction method for Fig. 2 is the same as Low et al. (2019) and Nayfach et al. (2021).

References:

Low, S. J., Džunková, M., Chaumeil, P. A., Parks, D. H. & Hugenholtz, P. Evaluation of a concatenated protein phylogeny for classification of tailed double-stranded DNA viruses belonging to the order Caudovirales. *Nat. Microbiol.* 4, 1306–1315 (2019).

Nayfach, S. et al. Metagenomic compendium of 189,680 DNA viruses from the human gut microbiome. *Nat. Microbiol.* 6, 960–970 (2021).

Of the 77 Caudoviricetes gene markers, 58 of them were detected in our datasets (please see those highlighted in yellow in Table 1 below). We have made available publicly the alignments of the marker genes with the sequences from this study at https://github.com/Dorothydu12/HK_BE_viromic.

We have clarified description of the methods for the tree construction in the main manuscript and Supplementary Note to indicate that the 77 Caudoviricetes gene markers were curated and we have provided the references for the methods we applied.

[Lines 460–463] “A maximum likelihood phylogenetic tree comprising Caudoviricetes vOTUs from this study and three publicly available virome datasets (SMGC, GOV, and MetaSUB) was constructed using 77 curated gene markers as described previously^{30,79}.

Supplementary Note [Lines 150–154] “A maximum likelihood phylogenetic tree comprising Caudoviricetes vOTUs from this study and the three publicly available virome datasets (SMGC, GOV, and MetaSUB) was constructed as described previously^{9,20}. Briefly, 77 curated Caudoviricetes markers were searched against the protein-coding gene sequences of the vOTUs using a profile hidden Markov model (HMM).”

[Lines 694–695, Figure 2 legend] “(a) A maximum likelihood phylogenetic tree of 87 species-level vOTUs derived from BEs.”

[Lines 696–697, Figure 2 legend] “(b) A maximum likelihood phylogenetic tree of 599 species-level vOTUs derived from BEs and other datasets (SMGC, GOV, and MetaSUB).”

Table 1. The 77 curated Caudoviricetes markers used for constructing the phylogenetic trees shown in Figure 2.

VOGid	Description	VOGid	Description
VOG00022	Integrase	VOG00478	Protein D14
VOG00024	DNA polymerase	VOG00524	SPBc2 prophage-derived uncharacterized protein YonJ
VOG00052	Head completion protein gp16	VOG00574	Gene 5 protein
VOG00053	DNA ligase	VOG00667	Baseplate wedge protein gp6
VOG00057	DNA primase	VOG00671	Terminase, large subunit gp19
VOG00061	Uncharacterized protein near the lysin gene (fragment)	VOG00687	Probable portal protein
VOG00080	Ribonucleoside-diphosphate reductase subunit alpha	VOG00688	Head completion protein Gp50
VOG00091	Deoxycytidylate deaminase	VOG00799	Portal protein gp20
VOG00094	Gene 69 protein	VOG00813	Minor tail protein Gp28
VOG00100	Ribonucleoside-diphosphate reductase large subunit	VOG00885	Neck protein gp13
VOG00108	ATP-dependent helicase 41	VOG00917	Hypothetical protein
VOG00110	Baseplate protein J	VOG00977	Uncharacterized 17.5-kDa protein in tk-vs intergenic region
VOG00122	Gene 22 protein	VOG01018	Minor tail protein Gp27
VOG00123	Exodeoxyribonuclease	VOG01027	ATP-dependent DNA helicase uvsW
VOG00143	Baseplate wedge protein gp25	VOG01065	Peptidoglycan hydrolase gp16
VOG00148	Hypothetical protein	VOG01068	Endolysin B
VOG00154	Gene 19 protein	VOG01103	Probable major capsid protein gp17
VOG00180	Major tail protein Gp23	VOG01124	Recombination protein uvsY
VOG00206	DNA-directed DNA polymerase	VOG01126	Spanin, inner membrane subunit
VOG00239	Sliding-clamp-loader gp44 subunit	VOG01137	Deoxynucleotide monophosphate kinase
VOG00246	Putative protein p41	VOG01286	Gene 54 protein
VOG00254	Ribonucleoside-diphosphate reductase subunit beta	VOG01550	Gene 15 protein
VOG00269	Minor capsid protein 10B	VOG01564	T7 RNA polymerase
VOG00282	Minor head protein GP7	VOG01638	Tail tubular protein gp11
VOG00318	Gene 31 protein	VOG01666	Recombination and repair protein
VOG00349	Antirepressor protein ant	VOG01872	Probable capsid assembly scaffolding protein
VOG00363	Tail sheath protein	VOG02305	Gene 32 protein
VOG00402	Exonuclease subunit 2	VOG02317	Gene 30 protein
VOG00413	Internal virion protein gp14	VOG02621	Single-stranded DNA-binding protein
VOG00439	Baseplate central spike complex protein gp27	VOG03123	Single-stranded DNA-binding protein
VOG00450	Prohead core protein protease	VOG03134	Portal protein

VOG00462	Portal protein	VOG03197	Ribonuclease H
VOG00629	Helix-destabilizing protein	VOG03229	PhoH-like protein
VOG03385	Tail completion protein gp15	VOG03444	Prohead protease
VOG03743	Probable flavin-dependent thymidylate synthase	VOG09266	Endonuclease I
VOG12699	Terminase, large subunit	VOG12984	RNA polymerase sigma factor
VOG17540	Minor tail protein	VOG18218	Gene 31 protein
VOG18677	Capsid assembly scaffolding protein	VOG20138	Tail tubular protein gp12
VOG21972	Gene 58 protein		

With regard to the ACR tree in Fig. 5, I still cannot understand what exactly was done to obtain this phylogeny.

The Acr tree shown in Fig. 5 was constructed using the 339 reference sequences from the Acr curated database PaCRISPR (Wang et al., 2020) (these sequences were experimentally validated by Dong et al., 2018, Marino et al., 2018, and Pawluk et al., 2016) and the 162 sequences from this study that were predicted by PaCRISPR. The reference and predicted Acr protein sequences were aligned using FAMSA (v.1.5.12) and trimmed using trimAl (v.1.4) to retain positions with < 50% gaps. The aligned sequences were used to construct a maximum-likelihood tree using FastTreeMP (v.2.1.11) with the auto model. The phylogenetic tree was visualized using iTOL.

We have provided additional details regarding the methods to construct the Acr tree in the Supplementary Note.

Supplementary Note [Lines 204–210] “The types of predicted representative Acr proteins were estimated using a PSI-BLAST search with the default parameters against the Acr curated database PaCRISPR³². Multiple sequence alignment between the predicted Acr proteins and the 339 reference Acr proteins obtained from the curated PaCRISPR database was generated using FAMSA (v.1.5.12)³⁶ and trimmed using trimAl (v.1.4)²¹ to retain positions with < 50% gaps. A maximum-likelihood phylogenetic tree was constructed with the aligned sequences using FastTreeMP (v.2.1.11) with the auto model²² and was visualized using iTOL (v.6; <https://itol.embl.de/>).”

We have also revised the legend of Fig. 5 to indicate the type of phylogenetic tree shown.

[Lines 728–730, Figure 5 legend] “(c) A maximum-likelihood phylogenetic tree of the 162 Acr proteins identified in this study and the 339 Acr proteins obtained from the curated PaCRISPR database⁴⁸.”

References:

- Dong, C. et al. Anti-CRISPRdb: a comprehensive online resource for anti-CRISPR proteins. *Nucleic Acids Res.* 46, D393–D398 (2018).
- Marino, N.D. et al. Discovery of widespread type I and type V CRISPR–Cas inhibitors. *Science* 362, 240–242 (2018).
- Pawluk, A. et al. Naturally occurring off-switches for CRISPR–Cas9. *Cell*, 167 1829–1838 (2016).
- Wang, J. et al. PaCRISPR: a server for predicting and visualizing anti-CRISPR proteins. *Nucleic Acids Res.* 48, W348–W357 (2020).

All the above apart, Reviewer 1 raises a variety of pertinent questions on phage biology, and I am not sure these are fully addressed in the revision.

Thank you for your comment. Based on the reply from Reviewer 1, we believe we have addressed all the questions related to phage biology from the last round of review.

Reviewer comments, third round –

Reviewer #3 (Remarks to the Author):

I found that in the second revision, the authors have addressed the remaining questions in a satisfactory manner. No further comments.